# Robust Generalized Schrödinger Bridge via Sparse Variational Gaussian Processes

**Minyoung Kim**
Samsung AI Center Cambridge, UK
mikim21@gmail.com

## Abstract

The famous Schrödinger bridge (SB) has gained renewed attention in the generative machine learning field these days for its successful applications in various areas including unsupervised image-to-image translation and particle crowd modeling. Recently, a promising algorithm dubbed GSBM was proposed to solve the generalized SB (GSB) problem, an extension of SB to deal with additional path constraints. Therein the SB is formulated as a minimal kinetic energy conditional flow matching problem, and an additional task-specific stage cost is introduced as the conditional stochastic optimal control (CondSOC) problem. The GSB is a new emerging problem with considerable room for research contributions, and we introduce a novel Gaussian process pinned marginal path posterior inference as a meaningful contribution in this area. Our main motivation is that the stage cost in GSBM, typically representing task-specific obstacles in the particle paths and other congestion penalties, can be potentially noisy and uncertain. Whereas the current GSBM approach regards this stage cost as a noise-free deterministic quantity in the CondSOC optimization, we instead model it as a stochastic quantity. Specifically, we impose a Gaussian process (GP) prior on the pinned marginal path, view the CondSOC objective as a (noisy) likelihood function, and infer the posterior path via sparse variational free-energy GP approximate inference. The main benefit is more flexible marginal path modeling that takes into account the uncertainty in the stage cost such as more realistic noisy observations. On some image-to-image translation and crowd navigation problems under noisy scenarios, we show that our proposed GP-based method yields more robust solutions than the original GSBM.

## 1 Introduction

The *bridge matching* problem is to find a stochastic/ordinary differential equation (SDE/ODE) that bridges two given distributions (Lipman et al., 2023; Albergo & Vanden-Eijnden, 2023; Neklyudov et al., 2023; Villani, 2009; Peyré & Cuturi, 2017). Its application areas are enormous, which also subsumes the recent generative modeling (e.g., conditional or unconditional image generation) (Sohl-Dickstein et al., 2015; Chen et al., 2018; Ho et al., 2020; Song et al., 2021). The famous Schrödinger bridge (SB) problem (Fortet, 1940; Kullback, 1968; Rüschendorf, 1995; De Bortoli et al., 2021; Vargas et al., 2021; Shi et al., 2023), a widely known problem for a century, is a special instance of the bridge problem. The SB problem has recently received renewed attention in the generative machine learning and related fields for its applications to unsupervised image-to-image translation (Kim et al., 2024), image inpainting (Wang et al., 2021), fluid modeling (Shi et al., 2023), molecular simulations (Noé et al., 2020), robot navigation (Liu et al., 2018), and more.

Among these recent trends and a large body of works, the generalization of SB (GSB), incorporating additional costs on marginal paths, has started drawing attention as an extension of the SB problem. One of the most promising methods called GSBM (Liu et al., 2024) was proposed, in which the SB is formulated as a minimal kinetic energy conditional flow matching problem (Lipman et al., 2023; Tong et al., 2023), and an additional task-specific stage cost is introduced in the conditional stochastic optimal control (CondSOC) problem. Incorporation of task-specific stage costs can give better guidance towards desired probability paths, or penalize prohibitive paths, which is desirable for realistic crowd navigation problems and latent-variable generative modeling problems.

GSBM opens up intriguing research possibilities, among which in this paper we are particularly interested in the CondSOC path optimization part. Although GSBM originally adopted deterministic

spline-based optimization for CondSOC, which can be effective in many existing domains, there are several rooms to improve. One is more flexible modeling. However, simply increasing polynomial degrees (e.g., cubic splines) is often prone to fail in practice mainly due to numerical issues, while path integral resampling (Kappen, 2005) for non-Gaussian path modeling is computationally expensive with some technical issues as discussed in Sec. 2. Another option for improvement is to take into account uncertainty in path modeling in order to lead to more robust GSB solutions.

The Bayesian Gaussian process (GP) approach can nicely fulfill these two desiderata. To this end, we view the CondSOC objective as the likelihood function, and impose a GP prior path measure to the mean and stdev paths. With sparse variational free energy formulation adapted from (Titsias, 2009), we can infer the posterior path measure effectively. Although this can incur additional computational complexity in the CondSOC optimization compared to GSBM's spline optimization, we are able to achieve significantly more flexible and robust solutions than GSBM by properly handling uncertainty in potentially noisy stage cost observation. Also, our variational GP inducing variables merely take up roughly twice as many as in GSBM's. Tested on several crowd navigation and image-to-image translation problems in noisy observation scenarios, we show that our GP-based approach is more robust to noise than GSBM by capturing uncertainty that may reside in the GSB problems.

## 2 BACKGROUND

While here we briefly provide background material required to understand our approach, for more comprehensive and detailed background, we refer the readers who are less familiar with this topic to our *unified bridge algorithm* (UBA) framework in Appendix A (also in a stand-alone article (Kim, 2025)). It not only gives introduction to the (conditional) flow matching (CFM) and Schrödinger bridge (SB) problems, but also offers a unified SDE framework that subsumes both CFM and SB into one.

**Bridge problems.** The SB is a special instance of the more general problem class called the *bridge problem* which aims to find an SDE (or sometimes an ODE) that bridges two given distributions. More specifically, given two distributions $\pi_0(\cdot)$ and $\pi_1(\cdot)$, the goal is to find an SDE, specifically the drift function $u_t(x)$, with a specified diffusion coefficient $\sigma$ ($\sigma = 0$ for ODE),

$$dx_t = u_t(x_t)dt + \sigma dW_t, \ x_0 \sim \pi_0(\cdot) \tag{1}$$

with the constraint that the state at time $t = 1$ conforms to the other target $\pi_1$, that is, $x_1 \sim \pi_1(\cdot)$. We can also find the reverse-time SDE for the bridge the other way around, i.e., starting from $\pi_1$ and landing at $\pi_0$. Once solved, the solution to the bridge problem can give us the ability to sample from one of $\pi_{\{0,1\}}$ given the samples from the other. For instance, in typical generative modeling, $\pi_0$ is usually a tractable density like Gaussian, while $\pi_1$ is a target distribution that we want to sample from. In the bridge problem, however, $\pi_0$ can also be an arbitrary distribution beyond tractable densities.

**Schrödinger bridge (SB)** has an additional constraint that the probability path measure should be the one that is closest to a given reference path measure (Appendix A.2.2). Denoting the path measure of (1) by $P^u$ and the reference measure by $P^{ref}$, the SB can be defined as the following optimization:

$$\min_u \ \mathrm{KL}(P^u || P^{ref}) \ \text{s.t.} \ P_0^u(x_0) = \pi_0(x_0), P_1^u(x_1) = \pi_1(x_1) \tag{2}$$

Without loss of generality, we use the Brownian SDE, $dx_t = \sigma dW_t$, as a reference measure $P^{ref}$. Several algorithms have been proposed to solve the SB: the traditional iterative filtering (IPF) algorithm, its extensions with modern deep neural nets, and mini-batch versions of (entropic) optimal transport couplings in CFMs. See details and related references in Appendix A.2.3 and A.2.1. The computational overheads and known drawbacks of these methods were addressed in the recent iterative projection method (IMF) also known as the DSBM algorithm (Shi et al., 2023) (Appendix A.2.4).

**Generalized Schrödinger bridge (GSB) problem.** The standard SB has been extended by certain state costs introduced to express application-specific preferences or penalties for state paths $\{x_t\}$ in SB (Chen et al., 2015; Liu et al., 2022; Chen, 2023; Liu et al., 2024). For instance, the state cost can encode particle interaction costs (Gaitonde et al., 2021), quantum potentials (Philippidis et al., 1979), or geometric costs (Liu et al., 2024). The GSB problem is what we mainly deal with in this paper, and we particularly focus on the recent GSB matching (GSBM) algorithm (Liu et al., 2024) for its superiority to its predecessors including (Liu et al., 2022) in terms of the quality of solutions.

For smooth exposition, we describe the GSBM algorithm within our UBA framework (Appendix A or (Kim, 2025)), in particular by showing how it can be extended from the minimal kinetic UBA form of the IMF/DSBM algorithm (Shi et al., 2023) described in Alg. 4. The following is a concise summary.

For the given state (or stage) cost $V_t(x)$, we repeat the following two steps (A and B) for a sufficient number of times to find a neural-net based SDE, $dx_t = v_\theta(t, x_t)dt + \sigma dW_t$:

$$\text{(Step-A)} \quad \arg\min_{\{\mu_t, \gamma_t\}_t} \int_0^1 \mathbb{E}_{P_t(x|x_0, x_1)}\left[\frac{1}{2}||\alpha_t(x|x_0, x_1)||^2 + V_t(x)\right]dt, \quad (x_0, x_1) \sim Q(x_0, x_1) \quad (3)$$

where $P_t(x|x_0, x_1) = \mathcal{N}(x; \mu_t, \gamma_t^2 I)$ is the pinned marginal at $t$ for coupling $(x_0, x_1)$ sampled from:

$$Q(x_0, x_1) := \begin{cases} \pi_0(x_0)\pi_1(x_1) & \text{initially (when the previous iterate } \theta^{old} \text{ is not available)} \\ P^{v_{\theta^{old}}}(x_0, x_1) & \text{driven by } dx_t = v_{\theta^{old}}(t, x_t)dt + \sigma dW_t, \ x_0 \sim \pi_0 \end{cases} \quad (4)$$

and $\alpha_t(x_t|x_0, x_1)$ is determined from $P_t(x|x_0, x_1)$ by the following formula ($\dot{f}_t$ is time derivative):

$$\alpha_t(x|x_0, x_1) = \dot{\mu}_t + a_t(x - \mu_t), \quad a_t = \dot{\gamma}_t/\gamma_t - \sigma^2/(2\gamma_t^2) \quad (5)$$

Once $P_t(x|x_0, x_1)$ is optimized for each coupling $(x_0, x_1)$, we update the neural net by:

$$\text{(Step-B)} \quad \theta \leftarrow \arg\min_\theta \mathbb{E}_{t, Q(x_0, x_1)P_t(x_t|x_0, x_1)}||\alpha_t(x_t|x_0, x_1) - v_\theta(t, x_t)||^2 \quad (6)$$

The rationale behind this stochastic optimal control (SOC) formulation is detailed in Appendix A.3.3. The appearance of the stage cost $V_t(x)$ in (3) is the only difference between DSBM and GSBM (i.e., it can be shown that the above procedure recovers DSBM exactly if $V = 0$). Following (Liu et al., 2024) we call the optimization (3) the *CondSOC* problem. In CondSOC, the stage cost $V$ is added to the kinetic energy term where the stage cost is task-specific, and expresses our problem-specific preferences or penalties (e.g., obstacles in crowd navigation or desired particle paths). In the GSBM algorithm, the mean $\mu_t$ and stdev $\gamma_t$ functions of the Gaussian[1] $P_t(x|x_0, x_1)$, i.e., the optimization variables of CondSOC, are parametrized as splines at some knot points.

## 3 GAUSSIAN PROCESS MODELING FOR PINNED MARGINAL PATHS

We denote the CondSOC objective in GSBM as $J(P_\bullet; V_\bullet)$ where $P_\bullet := \{P_t(\cdot|x_0, x_1)\}_t$ is the pinned marginal path, and $V_\bullet := \{V_t(\cdot)\}_t$ is the task-specific stage cost function. That is,

$$J(P_\bullet; V_\bullet) = \int_0^1 \mathbb{E}_{P_t(x_t|x_0, x_1)}\left[\frac{1}{2}||\alpha_t(x_t|x_0, x_1)||^2 + V_t(x_t)\right]dt \quad (7)$$

where $\alpha_t(x_t|x_0, x_1)$ is determined from $P_\bullet$ by (5). So, GSBM aims to find $P_\bullet$ by solving: $\arg\min_{P_\bullet} J(P_\bullet; V_\bullet)$. Note that in GSBM the pinned marginal path $P_\bullet$ is the *deterministic* optimization variables of the CondSOC, and they aim to find a point estimate. We instead treat $P_\bullet$ as a random variate, and place some prior distribution $\mathcal{P}_{prior}(P_\bullet)$. For instance, we can impose our preference of the SB's linear pinned marginal path over other less smooth ones through this prior. Then "$-\log \mathcal{P}_{prior}(P_\bullet)$" can be introduced as a regularizer to enforce our prior preference. With some balancing coefficient $\tau \ (\geq 0)$, what we call the *regularized-path GSBM* can be written as:

$$\arg\min_{P_\bullet} J(P_\bullet; V_\bullet) - \tau \log \mathcal{P}_{prior}(P_\bullet) \equiv \arg\max_{P_\bullet} \underbrace{\mathcal{P}_{prior}(P_\bullet)}_{\text{Prior}} \cdot \underbrace{\exp(-J(P_\bullet; V_\bullet)/\tau)}_{\text{Likelihood}} \quad (8)$$

where the equivalence comes from exponentiation. This can be seen as a maximum-a-posteriori (MAP) solution. But we go one step further to consider the posterior distribution,

$$\mathcal{P}_{post}(P_\bullet) \propto \mathcal{P}_{prior}(P_\bullet) \cdot \exp(-J(P_\bullet; V_\bullet)/\tau) \quad (9)$$

---

[1]Although in (Liu et al., 2024) they also proposed a non-Gaussian path modeling using the so-called path integral resampling, there are several technical issues, including: i) the related $\alpha_t(x|x_0, x_1)$ may not admit the non-Gaussian $P_t(x|x_0, x_1)$ as its marginal distribution, and ii) the importance sampling resampling procedure is computationally very overwhelming in practice. So, we adhere to the Gaussian path recipe as described here.

We specifically consider a Gaussian process (GP) prior for the mean and stdev functions ($\mu_\bullet :=\{\mu_t\}_t$ and $\gamma_\bullet := \{\gamma_t\}_t$). For computational tractability, we impose independent prior modeling with dimension-wise factorization for $\mu_t$. To ensure positivity of $\gamma_t$ we use the parameterization $\gamma_t = \sigma\sqrt{t(1-t)}\log(1+e^{\tilde{\gamma}_t})$, in which we impose a GP prior on the unconstrained $\tilde{\gamma}$. Specifically,

$$\mathcal{P}_{prior}(P_\bullet) = \mathcal{GP}(\mu_\bullet) \cdot \mathcal{GP}(\tilde{\gamma}_\bullet) = \prod_{j=1}^{d} \mathcal{GP}(\mu_\bullet^j; m_\bullet^{\mu,j}, k_{\bullet,\bullet}^{\mu,j}) \cdot \mathcal{GP}(\tilde{\gamma}_\bullet^j; m_\bullet^{\tilde{\gamma}}, k_{\bullet,\bullet}^{\tilde{\gamma}}) \tag{10}$$

where $m^{\{\mu,\tilde{\gamma}\}}$ and $k^{\{\mu,\tilde{\gamma}\}}$ are mean and covariance functions of GP for $\mu_\bullet$ and $\tilde{\gamma}_\bullet$. We use superscript $j$ for the $j$-th dimension (e.g., $\mu_\bullet^j$ is the $j$-th function of the $d$-dimensional vector function $\mu_\bullet$). A main benefit of this Bayesian path modeling is that by treating the potentially noisy CondSOC objective (e.g., resulting from noisy stage cost $V$) as a (stochastic) likelihood function, we can have a more robust path solution as a posterior process than simply a point estimate that fully trusts the objective $J$. Another benefit is that GP can lead to more natural and flexible path modeling.

Since we deal with $(x_0, x_1)$-pinned marginal paths, we need to ensure $\mu_0 = x_0$, $\mu_1 = x_1$, $\gamma_0 = \gamma_1 = 0$ for consistency with conditioning. For $\gamma_t$ we already meet these boundary conditions due to our construction. For $\mu_t$, we deal with the conditional process by conditioning $\mu_0^j = x_0^j$ and $\mu_1^j = x_1^j$ for each $j$, which is also a GP. More specifically, the conditional GP prior for $\mu_\bullet^j$ can be expressed as:

$$\mu_\bullet^j \mid (\mu_0^j = x_0^j, \mu_1^j = x_1^j) \sim \mathcal{GP}(M_\bullet^{\mu,j}, L_{\bullet,\bullet}^{\mu,j}) \quad \text{where} \tag{11}$$

$$M_t^{\mu,j} = m_t^{\mu,j} + k_{t,\{0,1\}}^{\mu,j}(k_{\{0,1\},\{0,1\}}^{\mu,j})^{-1}[x_0^j - m_0^j, x_1^j - m_1^j]^\top \tag{12}$$

$$L_{s,t}^{\mu,j} = k_{s,t}^{\mu,j} - k_{s,\{0,1\}}^{\mu,j}(k_{\{0,1\},\{0,1\}}^{\mu,j})^{-1}k_{\{0,1\},t}^{\mu,j} \tag{13}$$

where $k_{A,B}$ for index sets $A$, $B$ is defined to be the $(|A| \times |B|)$ matrix whose $(i,j)$ entry is $k_{A_i,B_j}$.

In all our experiments, the mean function for $\mu_t$ is chosen as the linear interpolation, $m_t^\mu = (1-t)x_0 + tx_1$, and the mean function of $m_t^{\tilde{\gamma}}$ is set to constant $\log(e-1)$, which in turn leads to $\sigma\sqrt{t(1-t)}$ for $\gamma_t$. This choice makes the prior mean coincide with the solution of DSBM in the original SB problem. For the covariance functions, we use the squared exponential kernel function (See ablation study on different kernel choices in Sec. 5.4), where the related kernel hyperparameters, denoted by $\eta$, are chosen by empirical Bayes (as discussed in the variational inference section below). For simplicity, we assume the same kernel function and hyperparameters for all dimensions of $\mu_t$.

### 3.1 Sparse Variational (Free-energy) GP Posterior Inference

Due to the intractability of the posterior $\mathcal{P}_{post}(P_\bullet)$ defined in (9), we adopt the sparse variational GP approximate inference (Titsias, 2009; Dezfouli & Bonilla, 2015; Matthews et al., 2016; Bauer et al., 2016) that is based on the variational free energy principle. First off, we consider the fully factorized GP as a tractable variational process family, specifically $\mathcal{Q}(P_\bullet) = \mathcal{Q}(\mu_\bullet) \cdot \mathcal{Q}(\tilde{\gamma}_\bullet) = \prod_{j=1}^{d} \mathcal{Q}(\mu_\bullet^j) \cdot \mathcal{Q}(\tilde{\gamma}_\bullet)$. Both $\mathcal{Q}(\mu_\bullet^j)$ and $\mathcal{Q}(\tilde{\gamma}_\bullet)$ are chosen to be Gaussian processes parametrized by the inducing-point processes. For exposition we focus on $\mathcal{Q}(\mu_\bullet^j)$ here since derivations for $\mathcal{Q}(\tilde{\gamma}_\bullet)$ are similar. For notational simplicity we will often drop superscript dependency on $\mu$ (and also $j$).

We choose $n$ inducing input (time) points $Z = (t_1, \ldots, t_n)$ with $0 < t_1 < \cdots < t_n < 1$. These pseudo inputs can be seen as representative time points for the posterior process in that knowing the values of $\mu_t$ at $t \in Z$ has decisive effects on inferring function values at the other input points. Further insights underlying the principle can be found in the nice survey (Quiñonero-Candela & Rasmussen, 2005). Although the inducing input points can be learned as well as model and variational parameters in the ELBO learning, we instead fix them as equal-spaced points in $[0, 1]$ for simplicity (Sec. 5.4 for ablation on $n$). Following the variational free-energy principle (Titsias, 2009), we define $\mathcal{Q}(\mu_\bullet)$ as:

$$\mathcal{Q}(\mu_\bullet) = \int \mathcal{Q}(\mu_Z)\mathcal{Q}(\mu_\bullet|\mu_Z)d\mu_Z \tag{14}$$

where $\mu_Z = [\mu_{t_1}, \ldots, \mu_{t_n}]^\top$, known as the *inducing variables*, is a $n$-dimensional vector of function values at the inducing inputs $Z$. The key idea is that we model only $\mathcal{Q}(\mu_Z)$ as a $n$-variate learnable Gaussian distribution while having $\mathcal{Q}(\mu_\bullet|\mu_Z)$ equal to $\mathcal{P}_{prior}(\mu_\bullet|\mu_Z)$, the conditional process

induced from the prior GP, which is also a GP. This not only helps us avoid modeling difficult conditional process $\mathcal{Q}(\mu_\bullet|\mu_Z)$, but also the integration in (14) becomes tractable leading to closed-form GP $\mathcal{Q}(\mu_\bullet)$. More specifically, we define:

$$\mathcal{Q}(\mu_Z^j) = \mathcal{N}(\mu_Z^j; C^{\mu,j}, S^{\mu,j}) \quad \text{for } j = 1, \ldots, d \tag{15}$$

where $C^{\mu,j}, S^{\mu,j}$ are $n$-dimensional variational parameter vectors by assuming a diagonal covariance. Note that when compared to GSBM's spline parametrization, assuming the number of spline knot points is the same as $n$ here, as they require $O(n)$ parameters to represent knot function and derivative values for each dimension $j$, we have the same complexity in terms of parametrization. And the conditional prior $\mathcal{P}_{prior}(\mu_\bullet|\mu_Z)$ is also a GP with mean and covariance functions written as:

$$\mathbb{E}[\mu_t] = M_t + L_{t,Z} L_{Z,Z}^{-1}(\mu_Z - M_Z), \quad \text{Cov}(\mu_s, \mu_t) = L_{s,t} - L_{s,Z} L_{Z,Z}^{-1} L_{Z,t} \tag{16}$$

where $M$ and $L$ are the mean and covariance functions of the boundary-conditioned prior GP (11).

Due to the Gaussianity of both terms, the integration (14) can be solved analytically, yielding a Gaussian process $\mathcal{Q}(\mu_\bullet)$ whose mean and covariance functions are:

$$\mathbb{E}_\mathcal{Q}[\mu_t] = M_t + L_{t,Z} L_{Z,Z}^{-1}(C - M_Z), \tag{17}$$

$$\text{Cov}_\mathcal{Q}(\mu_s, \mu_t) = L_{s,t} - L_{s,Z} L_{Z,Z}^{-1} S L_{Z,Z}^{-1} L_{Z,t} - L_{s,Z} L_{Z,Z}^{-1} L_{Z,t} \tag{18}$$

The variational posterior of the stdev function $\mathcal{Q}(\tilde{\gamma}_\bullet)$, following the same derivations, becomes a Gaussian process as well, with the following mean and covariances:

$$\mathbb{E}_\mathcal{Q}[\tilde{\gamma}_t] = m_t^{\tilde{\gamma}} + k_{t,Z'}^{\tilde{\gamma}}(k_{Z',Z'}^{\tilde{\gamma}})^{-1}(C^{\tilde{\gamma}} - m_{Z'}^{\tilde{\gamma}}), \tag{19}$$

$$\text{Cov}_\mathcal{Q}(\tilde{\gamma}_s, \tilde{\gamma}_t) = k_{s,t}^{\tilde{\gamma}} - k_{s,Z'}^{\tilde{\gamma}}(k_{Z',Z'}^{\tilde{\gamma}})^{-1} S^{\tilde{\gamma}}(k_{Z',Z'}^{\tilde{\gamma}})^{-1} k_{Z',t}^{\tilde{\gamma}} - k_{s,Z'}^{\tilde{\gamma}}(k_{Z',Z'}^{\tilde{\gamma}})^{-1} k_{Z',t}^{\tilde{\gamma}} \tag{20}$$

where $Z'$ is the inducing input times for $\gamma_t$, and $(C^{\tilde{\gamma}}, S^{\tilde{\gamma}})$ are the variational parameters for the inducing variables $\tilde{\gamma}_{Z'}$. Overall, $\Lambda := \{\{C^{\mu,j}, S^{\mu,j}\}_j, C^{\tilde{\gamma}}, S^{\tilde{\gamma}}\}$ constitute all our variational parameters.

**ELBO learning.** The ELBO objective can be derived from the non-negativity of the KL divergence between $\mathcal{Q}(P_\bullet)$ and $\mathcal{P}_{post}(P_\bullet)$ (Appendix B for derivations). We minimize the negative ELBO,

$$\min_{\Lambda, \eta, \tau} \mathbb{E}_{\mathcal{Q}(P_\bullet)}[J(P_\bullet; V_\bullet)/\tau] + \text{KL}(\mathcal{Q}(P_{Z,Z'})||\mathcal{P}_{prior}(P_{Z,Z'})) \tag{21}$$

with respect to the variational parameters $\Lambda$ as well as the model parameters $\eta$ (prior) and $\tau$ (likelihood). The variational-prior KL divergence, the second term in (21), is confined to $n$-dim Gaussians, and easy to compute in closed forms (Appendix B for details). The first term of (21) can be computed and optimized by reparametrized Monte-Carlo sampling using (17–20) (Kingma & Welling, 2014). For the reparametrized samples $P_\bullet$, that is, $(\mu_\bullet, \gamma_\bullet)$, we apply (5) to compute $J(P_\bullet; V_\bullet)$ where the time derivatives (e.g., $\dot{\mu}_t$) are computed[2] as $(\mu_{t+\Delta t} - \mu_t)/\Delta t$ utilizing the nearby covariance structure (say, $\text{Cov}_\mathcal{Q}(\mu_t, \mu_{t+\Delta t})$) and $\Delta t \to 0$. This ensures maintaining computational graphs for backpropagation. Since the (log-)evidence, a sole function of the model parameters $(\eta, \tau)$, is lower bounded by the ELBO where the gap is exactly $\text{KL}(\mathcal{Q}(P_\bullet)||\mathcal{P}_{post}(P_\bullet))$, minimizing (21) with respect to $\Lambda$ with fixed $(\eta, \tau)$ *guarantees* to reduce the gap, while minimizing it with respect to $(\eta, \tau)$ with fixed $\Lambda$ *potentially* improves the (log-)evidence. The latter optimization essentially amounts to performing empirical Bayes (evidence maximization), a principled way to do model selection. Our algorithm, dubbed **GP-GSBM**, is summarized in Alg. 1.

## 4 RELATED WORK

The core related works on Schrödinger bridge and generalized SB were discussed in Sec. 2 and Appendix A. Since our work is on application of Gaussian processes (Rasmussen & Williams, 2006) to the generalized Schrödinger bridge problem, here we briefly highlight two recent papers that applied GP to the bridge/flow problem, and discuss how they are different from ours.

**Stream-level flow-matching with GP** (Wei & Ma, 2025). This work is based on Tong et al. (2023)'s CFM framework whereas in our GP-GSBM the probability path GP is applied to the GSB problem.

---

[2]Since $\mu_\bullet$ is GP (17–18), $\dot{\mu}_\bullet$ is also GP whose mean and cov functions involve time derivatives of (17–18). But, we used approximation $(\mu_{t+\Delta t} - \mu_t)/\Delta t$ with small $\Delta t = 0.01$, which we found numerically more stable.

---

**Algorithm 1** Our GP-GSBM algorithm for robust generalized Schrödinger bridge matching.

---

**Input:** The end-point distributions $\pi_0$ and $\pi_1$ (i.e., samples from them) and the state cost $V_t(x)$.

**Repeat** until convergence or a sufficient number of times:

    0. Collect batch pairs $(x_0, x_1)$ sampled from the current estimate $Q(x_0, x_1)$ in (4).

    1. Solve the ELBO optimization in (21) to get variational parameters $\Lambda$ and model parameters $(\eta, \tau)$.

    2. Prepare a batch of $(x_0, x_1, t, x_t)$ and $\alpha_t(x_t|x_0, x_1)$ as follows:

        1) Sample $t \sim [0, 1]$ uniformly at random.

        2) Sample $(\mu_t, \gamma_t) \sim \mathcal{Q}(P_\bullet)$ using (17–20) with the learned parameters $\Lambda$ and $\eta$ for $\mathcal{Q}$.

        3) Sample $x_t \sim \mathcal{N}(x_t; \mu_t, \gamma_t^2 I)$.

        4) Compute $\alpha_t(x_t|x_0, x_1)$ in (5) using approximate time derivatives.

    3. Update the neural net of the SDE by running the following for $N_{nnet}$ iterations:

$$\theta \leftarrow \theta - \beta \nabla_\theta \mathbb{E}_{\text{batch}} ||\alpha_t(x_t|x_0, x_1) - v_\theta(t, x_t)||^2 \tag{22}$$

---

They have conditional GP derived by conditioning the prior GP on several fixed observed stream points. So this is the key difference. Recall that we take the sparse variational GP posterior inference while turning the CondSOC objective into a likelihood function. A limitation of their approach is that if there are some task-specific costs on the marginal paths (e.g., obstacles specified by the stage cost $V$), there is no way to penalize the GP path that passes through the obstacles systematically. That is, their GP framework may not be applicable to GSB problems unless one manually marks and provides the stream points to circumvent the obstacles, which is cumbersome and costly.

**Flow matching with GP prior** (Kollovieh et al., 2025). The setup and goal of this work are highly different from ours. They introduce GP as an additional prior structure on the distribution $\pi_0$ for generation of the time-series data $x_1$. Hence $x_0$ is the same dimensional time-series. Then they use either Langevin dynamics (Durmus & Éric Moulines, 2017) or posterior sampling (Dhariwal & Nichol, 2021; Kollovieh et al., 2023) in denoising diffusion models (Sohl-Dickstein et al., 2015; Ho et al., 2020; Song et al., 2021) with the mini-batch-OT-CFM learned from $\pi_0$. Hence this work is highly different from our approach of imposing a GP prior on the pinned marginal paths.

## 5 EXPERIMENTS

We test the performance of our GP-GSBM on several crowd navigation and image generation/translation problems. We mainly compare ours with GSBM (Liu et al., 2024) and DSBM (Shi et al., 2023): the former deals with the stage cost deterministically while the latter ignores the stage cost. The details of experimental setups including hyperparameters and neural net architectures are in Appendix C. Due to the lack of space, some parts of our experimental results are moved to Appendix D.

### 5.1 CROWD NAVIGATION PROBLEMS

We follow the problem setups and experimental settings from (Liu et al., 2022; 2024) where we focus on the geometric surface LiDAR problem (Sec. 5.1.1) and 2D obstacle problems (Sec. 5.1.2).

### 5.1.1 LIDAR GEOMETRIC SURFACE STAGE COSTS

We tackle the crowd navigation problem where the path cost is incurred through a complex geometric surface. Similar to the setup in (Liu et al., 2024), we deal with the surfaces observed through the LiDAR 3D scans of the Mt. Rainier (Legg & Anderson, 2013), where the surfaces form a manifold. We define the closeness to the manifold and the height as the path cost. Formally,

$$V_t(x) = ||\Pi_{\mathcal{M}}(x) - x||^2 + \exp(\Pi_{\mathcal{M}}(x)_{[z]}) \tag{23}$$

where $\Pi_{\mathcal{M}}(x)$ is the projection of the 3D point $x$ on to the manifold $\mathcal{M}$, which is done by tangent plane projection estimated by $x$'s $k$-nearest neighbor points ($k = 20$) on $\mathcal{M}$. The height of the projected point is the $z$-coordinate value $\Pi_{\mathcal{M}}(x)_{[z]}$.

This stage cost function encourages the navigation path to stay close to the surfaces and avoid high altitudes. Fig. 1 shows the sample paths from the competing models after training. We see that DSBM (leftmost), as expected, yields straight linear paths due to its ignorance of the stage cost, resulting in high CondSOC loss as summarized in Table 1. Our GP-GSBM successfully discovers the low

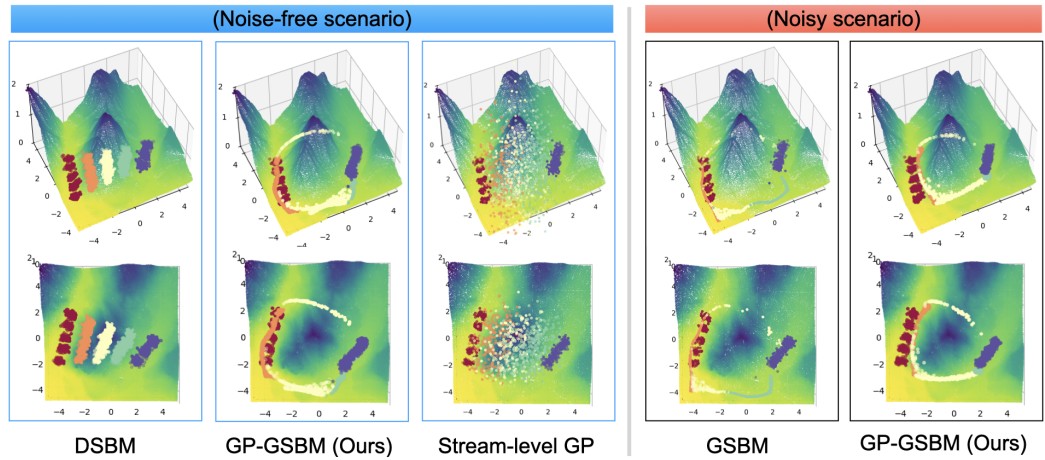

Figure 1: LiDAR path samples for different models in 3D (top rows) and 2D (bottom) views. Starting from the samples of $\pi_0$ shown as red points on the left, we generate samples for $\pi_1$ as blue points on the right. Path samples are visualized in 5 uniform time points from $t=0$ (red) to $t=1$ (blue).

Table 1: LiDAR results. CondSOC objective values for the learned models. Figures in parentheses are the Wasserstein distances between the true target $\pi_1$ and the learned ones.

|  | DSBM | GSBM | STREAM-LEVEL GP | GP-GSBM (OURS) |
|---|---|---|---|---|
| NOISE-FREE OBS. | $7747.0 \pm 76.4$ (0.04) | $6199.3 \pm 47.3$ (0.04) | $7012.6 \pm 61.9$ (0.15) | $\mathbf{5925.0} \pm 65.4$ (0.03) |
| NOISY OBS. | $12686.9 \pm 150.9$ (0.04) | $8506.1 \pm 65.6$ (0.04) | $12679.1 \pm 579.9$ (0.16) | $\mathbf{8300.0} \pm 67.6$ (0.04) |

altitude regions bypassing the saddle points as shown in the second column. GSBM (Liu et al., 2024) is also able to find out the desired two viable pathways similarly as ours (figure omitted here due to the similarity as ours). However, our GP-GSBM achieves even lower CondSOC cost perhaps due to more flexible pinned marginal path modeling than GSBM's deterministic spline optimization. Table 1 summarizes the CondSOC objective values for the learned models, averaged over 10 random runs.

**Noisy observation setup.** Since the projection to the manifold and reading the projected point can be a noisy process in practice, we mimic this noisy process by injecting random noise to the projected points. The results in Table 1 show that our GP-GSBM attains the lowest loss among the competing methods, signifying its benefit of robust path estimation through the Gaussian process inference under this noisy likelihood scenario. The visualized path samples in Fig. 1 (rightmost column) also shows that the desired two viable pathways are still well discovered by our model.

### 5.1.2 STAGE COSTS FROM OBSTACLES AND MEAN-FIELD INTERACTIONS

We next consider the crowd navigation problems with stage costs driven by obstacles and mean-field interactions. Similar to previous works (Liu et al., 2022; 2024), the stage cost is defined as:

$$V_t(x) = \lambda_{obs} L_{obs}(x) + \lambda_{ent} \log p_t(x) + \lambda_{cgst} \mathbb{E}_{x,x' \sim p_t}(1 + ||x - x'||^2)^{-1} \tag{24}$$

where $p_t(x)$ is the path marginal distributions obtained from the pinned marginals $p_t(x|x_0, x_1)$ and the coupling $Q(x_0, x_1)$. The obstacle cost $L_{obs}$ is defined as the distance between the state $x$ and the problem-specific obstacle. We may have the $\log p_t(x)$ term as the entropy cost encouraging diversified paths, and the congestion term to penalize densely packed particles along the paths. We test the competing models on two problem sets, Stunnel that has two ellipsoidal obstacles (Fig. 4) and GMM with three Gaussian blobs as obstacles (Fig. 5).

The learned $\mu_t$, $\gamma_t$ of the pinned marginals are shown in Fig. 4 and Fig. 5 while Table 2 reports the final CondSOC loss values averaged over 10 random runs. DSBM always finds a straight line, and incurs high CondSOC loss due to its ignorance of obstacles, i.e., the stage cost $V$. For Stunnel (Fig. 4), in the deterministic obstacle scenario, GSBM identifies an optimal path that circumvents the obstacles sharply while retaining minimal kinetic energy. Our GP-GSBM also finds a similar solution as GSBM's, and the posterior stdevs (red-shaded areas) are small in this case due to the deterministic nature of the problem. The model selection empirical Bayes finds small $\tau$ ($\approx 0.1$), implying that the model that puts more emphasis on the likelihood than the prior. This is reasonable and promising considering that the likelihood in this scenario is quite trustworthy.

Unlike in the original setup where the obstacles were always present, and the bridge matching algorithms can learn the stage cost function deterministically, we devise an uncertain setup by

Table 2: Stunnel and GMM results. CondSOC objective values for the learned models. Figures in parentheses are the Wasserstein distances between the true target $\pi_1$ and the learned ones.

|  |  | DSBM | GSBM | GP-GSBM (OURS) |
|---|---|---|---|---|
| STUNNEL | DETERMINISTIC | 18628.8 (0.02) | $492.94 \pm 1.13$ (0.03) | $\mathbf{488.78} \pm 1.07$ (0.02) |
|  | UNCERTAIN | 9549.2 (0.03) | $502.20 \pm 1.76$ (0.04) | $\mathbf{452.30} \pm 1.78$ (0.03) |
| GMM | DETERMINISTIC | 19824.2 (2.91) | $97.4 \pm 0.67$ (2.31) | $\mathbf{85.3} \pm 0.54$ (1.81) |
|  | UNCERTAIN | 13232.4 (2.96) | $101.6 \pm 1.02$ (2.83) | $\mathbf{89.2} \pm 0.62$ (2.19) |

regarding the presence of obstacles as a random process. Specifically we randomly turn on and off the obstacles with probability $p = 0.5$. This has the effect of turning the CondSoc objective function into an uncertain quantity, and we aim to judge which models are the most robust to such a noisy CondSOC objective function. We anticipate that our GP-GSBM will be more robust to GSBM (Liu et al., 2024) since we treat the CondSOC loss as a likelihood function, that is, a probabilistic model, rather than GSBM's deterministic treatment. As shown in Table 2, GSBM incurs even higher CondSOC loss than the deterministic case. GSBM is baffled in this case because it only optimizes the cost without taking into account the prior preference of straight line interpolation between $\pi_0$ and $\pi_1$, which is preferred for the obstacle absent cases. On the other hand, our GP-GSBM learns higher prior weight $\tau (\approx 1.0)$ from model selection, which encourages the marginal path to stay closer to the straight line interpolation (prior), preferable for obstacle absent cases. Furthermore, the uncertain stage cost is reflected in the stdev shading in our GP posterior where we observe even higher posterior uncertainty shown as larger red-shaded areas in Fig. 4 (the rightmost column).

## 5.2 UNPAIRED IMAGE-TO-IMAGE TRANSLATION

In this section we test the performance of our GP-GSBM on unsupervised image-to-image translation problem. Specifically, we deal with images of cats (regarded as $\pi_0$) and dogs (as $\pi_1$) from the AFHQ dataset (Choi et al., 2020). Although SB's optimal transport objective guides a bridge matching algorithm to learn a coupling between samples from $\pi_0$ and $\pi_1$ that is *optimal in the ambient pixel-space* L2 distance sense, this can often incur artifacts in the bridging paths. In other words, more intrinsic (latent) structure of the image manifolds cannot be taken into account.

To this end, following the latent space guidance proposed in (Liu et al., 2024), we incorporate the latent spherical-linear interpolated reconstruction error as the stage cost. More specifically,

$$V_t(x_t) = ||x_t - \text{dec}(z_t)||_1, \quad z_t = \text{slerp}(t, \text{enc}(x_0), \text{enc}(x_1)) \tag{25}$$

where $z = \text{enc}(x)$ and $x = \text{dec}(z)$ are the encoder and decoder of the pre-trained VAE model (Kingma & Welling, 2014). As claimed in (Liu et al., 2024), this stage cost helps preserving the underlying latent structures through semantically meaningful pinned marginal $P_t(x_t|x_0, x_1)$, yielding a faster training convergence and better couplings than models without the $V$ term such as DSBM (Shi et al., 2023). In addition to this observation, we further conjecture that the cost $V_t(x)$ itself is inherently noisy for several reasons, most notably that the learned VAE model may not be perfect in representing the true image manifold, and also the SLERP interpolation is only a proxy for the optimal latent paths. This is where our GP-GSBM can be especially beneficial for more robust path learning and bridging by handling such uncertainty in the stage cost in a principled Bayesian manner.

We trained our GP-GSBM with images of size $(64 \times 64)$ from cats and dogs, roughly $5K$ images from each. We first visualize how the prior and posterior pinned marginals $P_t(x|x_0, x_1)$ differ for dog $(t = 1)$ to cat $(t = 0)$ translation. After training the model, we pair the coupling of $(x_1, x_0)$ by running the trained (reverse-time) SDE model starting from $x_1 \sim \pi_1$, which is shown in Fig. 2(a) (more and magnified images in Fig. 6). From this coupling, we show the mean images from $P_t(x|x_0, x_1)$ for the prior and posterior (before and after the variational GP inference) in the middle. Visually clearly, the posterior exhibits semantically more meaningful path samples than the prior with less artifacts since the posterior takes into account the latent-preserving stage cost, forcing the samples to stay closer to the image manifold. In Fig. 2(b) (also Fig. 7), the SDE generation progress is shown with intermediate time steps. We also visualize the generated cat images in Fig. 2(c) (also Fig. 8). As a quantitative comparison, the FID scores of the generated cat images are reported in Table 3. Our GP-GSBM attains the lowest FID score, better than (deterministic) GSBM by capturing potential uncertainty/inaccuracy that may reside in the stage cost observation.

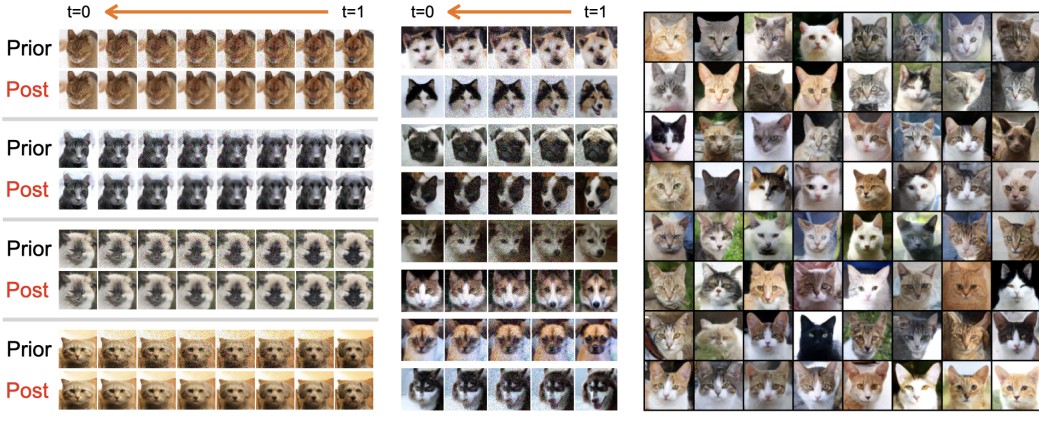

(a) Prior and posterior of $P_t(x \mid x_0, x_1)$     (b) Generation by SDE     (c) Generated images

Figure 2: Results of GP-GSBM on AFHQ dog ($t=1$, rightmost columns in (a), (b)) $\rightarrow$ cat ($t=0$, leftmost columns in (a), (b)) translation. See Appendix D.1 for more and enlarged images.

Table 3: AFHQ dog $\rightarrow$ cat image generation FID scores. Figures for DSBM (Shi et al., 2023) and GSBM (Liu et al., 2024) are excerpted from (Liu et al., 2024).

| DSBM | GSBM | STREAM-LEVEL GP | GP-GSBM (OURS) |
|---|---|---|---|
| 14.16 | 12.39 | 18.77 | **10.21** |

## 5.3 COMPARISON TO STREAM-LEVEL GP

As discussed in Sec. 4, our approach has several key differences from the conditional GP path modeling for CFM models in the recent work (Wei & Ma, 2025). Most notably, we regard GSBM's CondSOC objective as a likelihood function for GP posterior inference, whereas they used the linear interpolation velocity $\alpha_t(x_t|x_0, x_1) = \dot{x}_t$ as a conditional GP prior, thus difficult to incorporate task-specific stage costs. Here we perform some empirical tests to highlight the differences. We compare ours with the stream-level GP method on the LiDAR crowd navigation (Fig. 1, Table 1) and AFHQ image-to-image translation (Table 3). We adapted the stream-level GP within our UBA framework for these bridge matching problems. Overall the stream-level GP incurs high costs with lower distribution matching performance due to the ignorance of the stage costs. For additional results on a non-GSB problem, refer to Appendix D.2 for CIFAR-10 image generation experiments.

## 5.4 ABLATION STUDY AND RUNNING TIME (COMPUTATIONAL COMPLEXITY)

**Sensitivity to the number of inducing points ($n$) and the kernel function.** We used the default $n = 15$ (Stunnel) and $n = 30$ (LiDAR). To see how sensitive our GP-GSBM is to this hyperparameter, we vary $n$ in Fig. 3, which shows that the performance is not very sensitive to $n$

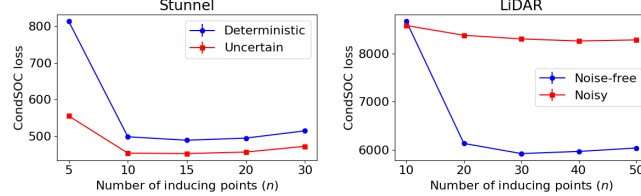

Figure 3: Sensitivity to the number of inducing points ($n$).

unless it is too small. We also test kernel types other than the default squared exponential kernel. As shown in Table 4 the polynomial kernel (degree selected from a grid of 1 to 10) slightly lags behind it.

**Running time.** In Appendix B.2 we analyzed asymptotic complexity of GP-GSBM compared to GSBM. Due to the GP modeling, our algorithm has extra cost of $O(n^2 d)$ (also $O(n^3)$ kernel inversion cost when we do model selection). Even though $n$ (the number of inducing points) is constant and usually no greater than 30 in practice, this may be computationally demanding especially for high-dimensional states. However, we suggested some workaround to reduce this cost in Appendix B.2. In this case, on LiDAR we got comparable time, GP-GSBM: 1.70 second per ELBO iteration, and

Table 4: CondSOC objectives for different kernels on Stunnel (left) and LiDAR (right).

| STUNNEL | SQUARED EXP. | POLYNOMIAL | | LIDAR | SQUARED EXP. | POLYNOMIAL |
|---|---|---|---|---|---|---|
| DETERMINISTIC | $488.78 \pm 1.07$ | $556.50 \pm 1.47$ | | NOISE-FREE | $5925.0 \pm 65.4$ | $6604.8 \pm 55.4$ |
| UNCERTAIN | $452.30 \pm 1.78$ | $435.37 \pm 9.31$ | | NOISY | $8300.0 \pm 67.6$ | $8625.2 \pm 65.8$ |

GSBM: 1.61 second per CondSOC iteration, when run on a single RTX-4090 GPU. We conduct more theoretical and empirical study to reduce further computational overhead as ongoing/future research.

## 6 CONCLUSION AND LIMITATION

We have proposed a novel Gaussian process approach to marginal path modeling for the generalized Schrödinger bridge problem. By imposing a GP prior on the pinned marginal path and viewing the CondSOC objective as a noisy likelihood function, the inferred posterior path in our model can lead to more flexible and robust solutions than the existing methods on problems under noisy observations and uncertainty. On several crowd navigation and image-to-image translation problems we have empirically demonstrated these benefits. A limitation of our current approach, however, is that it incurs higher computational complexity arising from GP posterior inference involving kernel inversion. Although we suggested some heuristic workarounds, further study on finding more principled solutions to reduce time complexity is needed, which we leave as future research.

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

# Appendix

## Table of Contents

## A  A UNIFIED FRAMEWORK FOR DIFFUSION BRIDGE PROBLEMS: FLOW MATCHING AND SCHRÖDINGER MATCHING INTO ONE

The *bridge problem* is to find a stochastic differential equation (SDE), or sometimes an ordinary differential equation (ODE), that bridges two given distributions. The application areas of the bridge problem are enormous, among which the recent generative modeling (e.g., conditional or unconditional image generation) is the most popular. Also the famous Schrödinger bridge problem, a widely known problem for a century, is a special instance of the bridge problem. Two most popular algorithms to tackle the bridge problems in the deep learning era are: *(conditional) flow matching* and *iterative fitting* algorithms, where the former confined to ODE solutions, and the latter specifically for the Schrödinger bridge problem. The main contribution of this section is in two folds: i) **We provide concise reviews of these algorithms with technical details to some extent**; ii) **We propose a novel unified perspective and framework that subsumes these seemingly unrelated algorithms (and their variants) into one**. In particular, we show that our unified framework can instantiate the Flow Matching (FM) algorithm, the (mini-batch) optimal transport FM algorithm, the (mini-batch) Schrödinger bridge FM algorithm, and the deep Schrödinger bridge matching (DSBM) algorithm as its special cases. We believe that this unified framework will be useful for viewing the bridge problems in a more general and flexible perspective, and in turn can help researchers and practitioners to develop new bridge algorithms in their fields.

### A.1  (DIFFUSION) BRIDGE PROBLEMS

The **diffusion bridge problem**, or simply the **bridge problem**, can be defined as follows.

• **Bridge problem.** Given two distributions $\pi_0(\cdot)$ and $\pi_1(\cdot)$ in $\mathbb{R}^d$, find an SDE, more specifically, find the drift function $u_t(x)$ where $u : \mathbb{R}[0,1] \times \mathbb{R}^d \to \mathbb{R}^d$ with a specified diffusion coefficient $\sigma$,

$$dx_t = u_t(x_t)dt + \sigma dW_t, \ x_0 \sim \pi_0(\cdot) \tag{26}$$

that yields $x_1 \sim \pi_1(\cdot)$. Here $\{W_t\}_t$ is the Wiener process or the Brownian motion. Alternatively one can aim to find a reverse-time SDE (or both). That is, find $\overline{u}_t(x)$ in

$$\overleftarrow{d}\,x_t = \overline{u}_t(x_t)dt + \sigma \overleftarrow{d}\,W_t, \ x_1 \sim \pi_1(\cdot) \tag{27}$$

that yields $x_0 \sim \pi_0(\cdot)$.

Note that if we specify $\sigma = 0$, then our goal is to find an ODE that bridges the two distributions $\pi_0(\cdot)$ and $\pi_1(\cdot)$. Once solved, the solution to the bridge problem can give us the ability to sample from one of the $\pi_{\{0,1\}}$ given the samples from the other, simply by integrating the learned SDE. The application areas of the bridge problem are enormous, among which the generative modeling (e.g., conditional or unconditional image generation) is the most popular. For instance, in typical generative modeling, $\pi_0$ is usually a tractable density like Gaussian, while $\pi_1$ is a target distribution that we want to sample from. In the bridge problem, however, $\pi_0$ can also be an arbitrary distribution beyond tractable densities like Gaussians, and we do not make any particular assumption on $\pi_0$ and $\pi_1$ as long as we have samples from the two distributions.

• **Two instances of the bridge problem.** There are two interesting special instances of the bridge problem: the **Schrödinger bridge problem** and the **ODE bridge problem**.

- **ODE bridge problem.** We strictly restrict ourselves to ODEs, i.e., $\sigma = 0$.
- **Schrödinger bridge problem.** With $\sigma > 0$, there is an additional constraint that the path measure of the SDE (26), denoted by $P^u$, is closest to a given reference SDE path measure $P^{ref}$. That is,

$$\min_u \text{ KL}(P^u || P^{ref}) \text{ s.t. } P_0^u(x_0) = \pi_0(x_0), P_1^u(x_1) = \pi_1(x_1) \tag{28}$$

where $P_t$ of a path measure $P$ indicates the marginal distribution at time $t$. In this case to have finite KL divergence, $\sigma$ of $P^u$ has to be set equal to $\sigma_{ref}$ of $P^{ref}$, i.e., $\sigma = \sigma_{ref}$, (from the Girsanov theorem).

## A.2 Flow Matching and Schrödinger Bridge Matching Algorithms

Among several existing algorithms that aim to solve the bridge problems, here we focus on two recent matching algorithms: *(conditional) flow matching* (Lipman et al., 2023; Tong et al., 2023; Albergo & Vanden-Eijnden, 2023; Liu et al., 2023) and *Schrödinger bridge matching* algorithms (De Bortoli et al., 2021; Vargas et al., 2021; Shi et al., 2023). These algorithms were developed independently: the former aimed to solve the ODE bridge problem while the latter the Schrödinger bridge problem. In this section we review the algorithms focusing mainly on the key ideas with some technical details, but being mathematically less rigorous for better readability.

In Sec. A.3, we propose a novel unified framework that subsumes these two seemingly unrelated algorithms and their variants into one.

### A.2.1 (Conditional) Flow Matching for ODE Bridge Problems

The Flow Matching (FM) (Lipman et al., 2023) or its extension Conditional Flow Matching (CFM) (Tong et al., 2023) is one promising way to solve the ODE bridge problem. The key idea of the FM is quite intuitive. We first design some marginal distribution path $\{P_t(x_t)\}_t$ with the boundary conditions $P_0 = \pi_0$, $P_1 = \pi_1$. We then derive the ODE $dx_t = u_t(x_t)dt$ that yields $\{P_t(x_t)\}_t$ as its marginal distributions. The drift $u_t(x_t)$ is approximated by a neural network $v_\theta(t, x_t)$ with parameters $\theta$ by solving:

$$\min_\theta \mathbb{E}_{t,x_t \sim P_t} ||u_t(x_t) - v_\theta(t, x_t)||^2 \tag{29}$$

Once solved, we can generate samples from $\pi_1$ (or $\pi_0$) approximately by simulating: $dx_t = v_\theta(t, x_t)dt$, $x_0 \sim \pi_0$ (resp., $x_1 \sim \pi_1$). However, one of the main limitations of this strategy is that designing the marginal path $\{P_t(x_t)\}_t$ satisfying the boundary condition is often difficult. And it is this issue that motivated the CFM.

• **Conditional Flow Matching (CFM).** To make the path design easier, we introduce some latent random variable $z$ to condition $x_t$. Although CFM derivations hold regardless of the choice of $z$, it is typically chosen as the terminal random variates $z = (x_0, x_1)$, and we will follow this practice and notation. Specifically, in CFM we design the so-called *pinned marginal path* $\{P(x_t|x_0, x_1)\}_t$ and the *coupling distribution* $Q(x_0, x_1)$ subject to the condition $P_0(x_0) = \pi_0(x_0)$, $P_1(x_1) = \pi_1(x_1)$ where $P_t(x_t)$ is defined as

$$P_t(x_t) := \int P_t(x_t|x_0, x_1)Q(x_0, x_1)d(x_0, x_1) \tag{30}$$

Then we derive the ODE $dx_t = u_t(x_t|x_0, x_1)dt$ that yields $\{P(x_t|x_0, x_1)\}_t$ as its marginal distributions for each $(x_0, x_1)$, which admits a closed form if $\{P(x_t|x_0, x_1)\}_t$ are Gaussians (Lipman et al., 2023). We then approximate $\mathbb{E}[u_t(x_t|x_0, x_1)|x_t]$, the conditional expectation derived from the joint $P(x_t|x_0, x_1)Q(x_0, x_1)$, by a neural network $v_\theta(t, x_t)$ by solving the following optimization:

$$\min_\theta \mathbb{E} ||u_t(x_t|x_0, x_1) - v_\theta(t, x_t)||^2 \tag{31}$$

where the expectation is taken with respect to the joint $P(x_t|x_0, x_1)Q(x_0, x_1)$ and uniform $t$. Surprisingly, it can be shown (Tong et al., 2023) that the gradient of the objective in (31) coincides with that in (29) for $u_t(x_t)$ defined as:

$$u_t(x) = \frac{1}{P_t(x_t)}\mathbb{E}_{Q(x_0, x_1)}[u_t(x_t|x_0, x_1)P_t(x_t|x_0, x_1)] \tag{32}$$

hence sharing the same training dynamics as (29). Therefore the optimal $v_\theta(t, x_t)$ of (31) is a good estimate for $u_t(x_t)$. Since the ODE $dx_t = u_t(x_t)dt$ admits $\{P_t(x_t)\}_t$ as its marginal distributions, so does $dx_t = v_\theta(t, x_t)dt$ approximately.

Many existing flow matching variants including FM (Lipman et al., 2023), Stochastic Interpolation (Albergo & Vanden-Eijnden, 2023), and Rectified FM (Liu et al., 2023) can be viewed as special instances of this CFM framework. For instance, these models can be realized by having a straight line (linear interpolation) pinned marginal path (33) with vanishing variance while one boundary (e.g., $\pi_0$) is fixed as standard normal $\mathcal{N}(0, I)$.

• **Limitations of CFM.** A reasonable choice for the coupling distribution $Q(x_0, x_1)$ is the Optimal Transport (OT) or the entropic OT between $\pi_0$ and $\pi_1$. The pinned marginal $P_t(x_t|x_0, x_1)$ can be chosen as a Gaussian with the linear interpolation between $x_0$ and $x_1$ as its mean, more specifically,

$$P_t(x_t|x_0, x_1) = \mathcal{N}(tx_1 + (1-t)x_0, \beta_t^2 I) \tag{33}$$

for some scheduled variances $\beta_t^2$. When the combination of the entropic OT $Q(x_0, x_1)$ and $\beta_t = \sigma_{ref}\sqrt{t(1-t)}$ is used, it can be shown that the marginals $\{P_t(x_t)\}_t$ coincide with the marginals of the Schrödinger bridge with the Brownian motion reference $P^{ref} : dx_t = \sigma_{ref}dW_t$. However, the main limitations of CFM (Tong et al., 2023) are: i) CFM solutions are confined to ODEs, hence unable to find the optimal SDE solution to general bridge problems including the Schrödinger bridge problem; ii) CFM itself does not provide a recipe about how to solve the entropic OT problem exactly – what is called SB-CFM proposed in (Tong et al., 2023) only approximates it with the Sinkhorn-Knopp solution for minibatch data, which is usually substantially different from the population entropic OT solution, that is, the solution to the Schrödinger bridge problem (See the SB static view in Sec. A.2.2).

### A.2.2 SCHRÖDINGER BRIDGE PROBLEM

The Schrödinger bridge problem can be defined as (28) where we assume a zero-drift Brownian SDE with diffusion coefficient $\sigma_{ref}$ for the reference path measure $P^{ref}$. That is,

$$P^{ref} : dx_t = \sigma_{ref}dW_t, \ x_0 \sim \pi_0(\cdot) \tag{34}$$

We denote by $P^{SB}$ the Schrödinger bridge path measure, i.e., the solution to (28). In the literature, there are two well-known views for $P^{SB}$: the *static view* and the *optimal control* view.

The static view has a direct link to the entropic optimal transport (EOT) solution, more specifically

$$P^{SB}(\{x_t\}_{t \in [0,1]}) = P^{EOT}(x_0, x_1) \cdot P^{ref}(\{x_t\}_{t \in (0,1)}|x_0, x_1) \tag{35}$$

where $P^{EOT}(x_0, x_1)$ is the EOT joint distribution solution with the negative entropy regularizing coefficient $2\sigma_{ref}^2$. More formally,

$$P^{EOT}(x_0, x_1) = \arg\min_{P(x_0, x_1)} \mathbb{E}_{P(x_0, x_1)}||x_0 - x_1||^2 - 2\sigma_{ref}^2 \mathbb{H}(P(x_0, x_1)) \tag{36}$$

$$\text{s.t.} \quad P(x_0) = \pi_0(x_0), \ P(x_1) = \pi_1(x_1) \tag{37}$$

where $\mathbb{H}$ indicates the Shannon entropy. We call $P^{ref}(\cdot|x_0, x_1)$ the *pinned reference process*, which admits a closed-form Gaussian expression for the specific choice (34). Although the product form (35), i.e., the product of the boundary joint distribution and the pinned path measure, does not in general become Markovian (e.g., Itô SDE representable), the Schrödinger bridge is a well-known exception where there exists a unique SDE that yields $P^{SB}$ as its path measure.

Alternatively, it is not difficult to derive an optimal control formulation for the Schrödinger bridge problem. Specifically, $P^{SB}$ can be described by the SDE that has the minimum kinetic energy among those that satisfy the bridge constraint. Letting

$$P^v : dx_t = v_t(x_t)dt + \sigma_{ref}dW_t, \ x_0 \sim \pi_0(\cdot) \tag{38}$$

we have $P^{SB} = P^{v^*}$ where $v^*$ is the minimizer of the following problem:

$$\min_v \mathbb{E}_{P^v}\left[\int_0^1 \frac{1}{2\sigma_{ref}^2}||v_t(x_t)||^2 dt\right] \text{ s.t. } P_1^v(x_1) = \pi_1(x_1) \tag{39}$$

Next we summarize two recent algorithms that solve the Schrödinger bridge problem exactly (at least in theory): Iterative Proportional Filtering (IPF) and Iterative Markovian Fitting (IMF).

### A.2.3 ITERATIVE PROPORTIONAL FILTERING (IPF)

IPF aims to solve the Schrödinger bridge problem (28) by alternating the forward and reverse half bridge (HB) problems until convergence. More specifically, with initial $P^0 = P^{ref}$, we solve the followings for $n = 1, 2, \ldots$

$$\text{(Reverse HB)} \quad P^{2n-1} = \arg\min_P \text{ KL}(P||P^{2n-2}) \text{ s.t. } P_1(x_1) = \pi_1(x_1) \tag{40}$$

$$\text{(Forward HB)} \quad P^{2n} = \arg\min_P \text{ KL}(P||P^{2n-1}) \text{ s.t. } P_0(x_0) = \pi_0(x_0) \tag{41}$$

It can be shown that $\lim_{n\to\infty} P^n \to P^{SB}$ (Fortet, 1940; Kullback, 1968; Rüschendorf, 1995). It is not difficult to show that the optimal solution of (40) or (41) can be attained by time-reversing the SDE of the previous iteration. This fact was exploited recently in (De Bortoli et al., 2021; Vargas et al., 2021) to yield neural-network based IPF algorithms where the score $\nabla \log P^n(x)$ that appears in time reversal is estimated either by regression estimation (De Bortoli et al., 2021) or maximum likelihood estimation (Vargas et al., 2021). However, the main drawback of these IPF algorithms is that they are simulation-based methods, thus very expensive to train.

### A.2.4 ITERATIVE MARKOVIAN FITTING (IMF)

Recently in (Shi et al., 2023), the concept of path measure projection was introduced, specifically the Markovian and reciprocal projections that preserve the boundary marginals of the path measure. This idea was developed into a novel matching algorithm called the *iterative Markovian fitting* (IMF) that alternates applying the two projections starting from the initial path measure. Not only is it shown to converge to $P^{SB}$, but the algorithm is computationally more efficient than IPF without relying on simulation-based learning. A practical version of the algorithm is dubbed *Deep Schrödinger Bridge Matching* (DSBM).

We begin with discussing the two projections.

• **Reciprocal projection.** They define the *reciprocal class* of path measures to be the set of path measures that admit $P^{ref}(\cdot|x_0, x_1)$ as their pinned conditional path measures. That is, the *reciprocal class* $\mathcal{R}$ is defined as:

$$\mathcal{R} = \{P : P(x_0, x_1)P^{ref}(\cdot|x_0, x_1)\} \tag{42}$$

The *reciprocal projection* of a path measure $P$, denoted by $\Pi_{\mathcal{R}}(P)$, is defined as the path measure in the reciprocal class that is closest to $P$ in the KL divergence sense. Formally,

$$\Pi_{\mathcal{R}}(P) = \arg\min_{R\in\mathcal{R}} \text{KL}(P||R) = P(x_0, x_1)P^{ref}(\cdot|x_0, x_1) \tag{43}$$

where the latter equality can be easily derived from the KL decomposition property. So it basically says that the reciprocal projection of $P$ is simply done by replacing $P(\cdot|x_0, x_1)$ by that of $P^{Ref}$ while keeping the coupling $P(x_0, x_1)$.

• **Markovian projection.** They also define the *Markovian class* as the set of any SDE-representable path measures with diffusion coefficient $\sigma_{ref}$. That is,

$$\mathcal{M} = \{P : dx_t = g_t(x_t)dt + \sigma_{ref}dW_t \text{ for any vector field } g\} \tag{44}$$

The *Markovian projection* of a path measure $P$, denoted by $\Pi_{\mathcal{M}}(P)$, is defined similarly as the path measure in the Markovian class that is closest to $P$ in the KL divergence sense,

$$\Pi_{\mathcal{M}}(P) = \arg\min_{M\in\mathcal{M}} \text{KL}(P||M) \tag{45}$$

In (Shi et al., 2023) (Proposition 2 therein), it was shown that $\Pi_{\mathcal{M}}(P)$ can be expressed succinctly for reciprocal path measures $P$. Specifically, for $P \in \mathcal{R}$, we have $\Pi_{\mathcal{M}}(P) = P^{v^*}$ where $P^{v^*}$ is described by the SDE: $dx_t = v_t^*(x_t)dt + \sigma_{ref}dW_t, \ x_0 \sim P(x_0)$ where

$$v_t^*(x_t) = \mathbb{E}_{P(x_1|x_t)}\left[\sigma_{ref}^2 \nabla_{x_t} \log P^{ref}(x_1|x_t)\right] = \mathbb{E}_{P(x_1|x_t)}\left[\frac{x_1 - x_t}{1 - t}\right] \tag{46}$$

where the latter equality comes immediately from the closed-form $P^{ref}(x_1|x_t) = \mathcal{N}(x_t, \sigma_{ref}^2(1-t)I)$. Also it was shown that the marginals are preserved after the projection, that is, $P_t^{v^*}(\cdot) = P_t(\cdot)$

for all $t \in [0, 1]$. This means that applying any number of Markovian (and also reciprocal) projections to a path measure $P$ always preserves the boundary marginals $P_0(x_0)$ and $P_1(x_1)$. And this is one of the key theoretical underpinnings of their algorithms called IMF and its practical version DSBM (details below) to solve the Schrödinger bridge problem.

• **IMF and DSBM algorithms.** Conceptually the IMF algorithm can be seen as a successive alternating application of the Markovian and reciprocal projections, starting from any initial path measure $P^0$ that satisfies $P^0(x_0) = \pi_0(x_0)$ and $P^0(x_1) = \pi_1(x_1)$ (e.g., $P^0 = \pi_0(x_0)\pi_1(x_1)P^{ref}(\cdot|x_0, x_1)$ is a typical choice). That is, for $n = 1, 2, \ldots$

$$P^{2n-1} = \Pi_{\mathcal{M}}(P^{2n-2}), \quad P^{2n} = \Pi_{\mathcal{R}}(P^{2n-1}) \tag{47}$$

Not only do all $\{P^n\}_{n \geq 0}$ meet the boundary conditions (i.e., $P_0^n = \pi_0$, $P_1^n = \pi_1$), it can be also shown that they keep getting closer to $P^{SB}$, and converge to $P^{SB}$ (i.e., $\mathrm{KL}(P^{n+1}||P^{SB}) \leq \mathrm{KL}(P^n||P^{SB})$ and $\lim_{n \to \infty} P^n = P^{SB}$) (Shi et al., 2023) (Proposition 7 and Theorem 8 therein). The reciprocal projection is straightforward as it only requires sampling from the pinned process $P^{ref}(\cdot|x_0, x_1)$ that is done by running $dx_t = \frac{x_1 - x_t}{1-t}dt + \sigma_{ref}dW_t$ with $(x_0, x_1)$ taken from the previous path measure. However, the Markovian projection involves the difficult $P(x_1|x_t)$ in (46) from the previous path measure $P$. To circumvent $P(x_1|x_t)$, they used the regression theorem by introducing a neural network $v_\theta(t, x)$ to approximate $v_t^*(x)$ and optimizing the following:

$$\arg\min_\theta \int_0^1 \mathbb{E}_{P(x_t, x_1)}\big|\big|v_\theta(t, x_t) - \sigma_{ref}^2 \nabla_{x_t} \log P^{ref}(x_1|x_t)\big|\big|^2 dt \tag{48}$$

where now the cached samples $(x_1, x_t)$ from the previous path measure $P$ can be used to solve (48). Although theoretically $v_{\theta^*}(t, x) = v_t^*(x)$ with ideally rich neural network capacity and perfect optimization, in practice due to the neural network approximation error, the boundary condition is not satisfied, i.e., $P_1^{2n-1} \neq \pi_1$. Hence to mitigate the issue, they proposed IMF's practical version, called the Diffusion Schrodinger Bridge Matching (DSBM) algorithm (Shi et al., 2023). The idea is to do Markovian projections with both forward and reverse-time SDEs in an alternating fashion where the former starts from $\pi_0$ and the latter from $\pi_1$, which was shown to mitigate the boundary condition issue.

## A.3 A UNIFIED FRAMEWORK FOR DIFFUSION BRIDGE MATCHING PROBLEMS

Our proposed unified framework is described in Alg. 2. It can be seen as an extension of the CFM algorithm (Tong et al., 2023) where the difference is that we consider the SDE bridge instead of the ODE bridge (i.e., the diffusion term in step 2). But this difference is crucial, as will be shown, allowing us to resolve the limitations of the CFM discussed in Sec. A.2.1. It also makes the framework general enough to subsume the IMF/DSBM algorithm for the Schrödinger bridge problem and various ODE bridge algorithms as special cases. We also emphasize that even though this small change of adding the diffusion term in step 2 may look minor, its theoretical consequence, specifically our theoretical result in Theorem A.1, has rarely been studied in the literature by far.

We call the unified framework *Unified Bridge Algorithm* (UBA for short). Note that UBA described in Alg. 2 can deal with both ODE and SDE bridge problems, and if the diffusion coefficient $\sigma$ vanishes, it reduces to CFM for ODE bridge. Similarly as CFM, under the assumption of rich enough neural network functional capacity and perfect optimization solutions, our framework *guarantees* to solve the bridge problem. More formally, we have the following theorem.

**Theorem A.1** (Our Unified Bridge Algorithm (UBA) solves the bridge problem). *If the neural network $v_\theta(t, x)$ functional space is rich enough to approximate any function arbitrarily closely, and if the optimization in step 3 can be solved perfectly, then each iteration of going through steps 1–3 in Alg. 2 ensures that $dx_t = v_\theta(t, x_t)dt + \sigma dW_t$, $x_0 \sim \pi_0(\cdot)$ (after the optimization in step 3) admits $\{P_t(x_t)\}_t$ of (49) as its marginal distributions.*

The proof can be found in Sec. A.4. The theorem says that after each iteration of going through steps 1–3, it is always guaranteed that the current SDE admits $\{P_t(x_t)\}_t$ defined in step 1 as marginal distributions. Since $P_0(\cdot) = \pi_0(\cdot)$ and $P_1(\cdot) = \pi_1(\cdot)$, the bridge problem is solved. Depending on the design choice, one can have just one iteration to solve the bridge problem. Under certain choices, however, it might be necessary to run the iterations many times to find the desired bridge solutions

---

**Algorithm 2** Our Unified Bridge Algorithm (UBA) for bridge problems.

---

**Input:** The end-point distributions $\pi_0$ and $\pi_1$ (i.e., samples from them).

**Repeat** until convergence or a sufficient number of times:

  1. Choose a pinned marginal path $\{P_t(x|x_0, x_1)\}_t$ and a coupling distribution $Q(x_0, x_1)$ such that $P_0(\cdot) = \pi_0(\cdot)$ and $P_1(\cdot) = \pi_1(\cdot)$ where

$$P_t(x_t) := \int P_t(x_t|x_0, x_1)Q(x_0, x_1)d(x_0, x_1) \tag{49}$$

  2. Choose $\sigma \geq 0$, and find $u_t(x|x_0, x_1)$ such that the SDE

$$dx_t = u_t(x_t|x_0, x_1)dt + \sigma dW_t \tag{50}$$

  admits $\{P_t(x|x_0, x_1)\}_t$ as its marginals. (Note: many possible choices for $\sigma$ and $u_t(x|x_0, x_1)$)
  3. Solve the following optimization problem with respect to the neural network $v_\theta(t, x)$:

$$\min_\theta \; \mathbb{E}_{t,Q(x_0,x_1)P_t(x_t|x_0,x_1)}||u_t(x_t|x_0, x_1) - v_\theta(t, x_t)||^2 \tag{51}$$

**Return:** The learned SDE $dx_t = v_\theta(t, x_t)dt + \sigma dW_t$ as the bridge problem solution.

---

(e.g., mini-batch OT-CFM (Tong et al., 2023) and the IMF/DSBM Schrödinger bridge matching algorithm (Shi et al., 2023) as we illustrate in Sec. A.3.1 and Sec. A.3.3, respectively).

In the subsequent sections, we illustrate how several popular ODE bridge and Schrödinger bridge algorithms can be instantiated as special cases of our UBA framework.

### A.3.1 A SPECIAL CASE: (MINI-BATCH) OPTIMAL TRANSPORT CFM (TONG ET AL., 2023)

Within our general Unified Bridge Algorithm (UBA) framework (Alg. 2), we select $P_t(x|x_0, x_1)$, $Q(x_0, x_1)$ and $u_t(x|x_0, x_1)$ as follows. First in step 1,

$$P_t(x_t|x_0, x_1) = \mathcal{N}(x_t; (1 - t)x_0 + tx_1, \sigma_{min}^2 I) \tag{52}$$

$$Q(x_0, x_1) = P^{mOT}(x_0, x_1) := \sum_{i \in B_0} \sum_{j \in B_1} \delta(x_0 = x_0^i)\delta(x_1 = x_1^j)p_{ij}^{mOT} \tag{53}$$

where $\sigma_{min} \to 0$, and $(\{x_0^i\}_{i \in B_0}, \{x_1^j\}_{j \in B_1})$ is the mini-batch data, and $\{p_{ij}^{mOT}\}_{ij}$ is a $(|B_0| \times |B_1|)$ mini-batch OT solution matrix learned with the mini-batch as training data. That is,

$$p^{mOT} = \arg\min_p \sum_{i,j} p_{ij}||x_0^i - x_1^j||^2 \quad \text{s.t.} \quad \sum_{j \in B_1} p_{ij} = \frac{1}{|B_0|}, \sum_{i \in B_0} p_{ij} = \frac{1}{|B_1|} \tag{54}$$

It is worth mentioning that $\sigma_{min} \to 0$ is required to have boundary consistency for $P_t(x_t|x_0, x_1)$ at $t = 0$ and $t = 1$. Note also that $P_t(x|x_0, x_1)$ is always fixed over iterations while $Q(x_0, x_1)$ varies over iterations depending on the mini-batch data sampled. Note that in our UBA framework, each iteration allows for different choices of $P_t(x|x_0, x_1)$ and $Q(x_0, x_1)$.

In step 2, we choose $\sigma = 0$, and define $u_t(x_t|x_0, x_1)$ to be a constant (independent on $t$) straight line vector from $x_0$ to $x_1$, i.e.,

$$u_t(x_t|x_0, x_1) = x_1 - x_0 \tag{55}$$

which can be shown to make the ODE $dx_t = u_t(x_t|x_0, x_1)dt$ admit $P_t(x_t|x_0, x_1)$ as its marginal distributions (Tong et al., 2023).

The above choices precisely yield the (mini-batch) optimal transport CFM (OT-CFM) introduced in (Tong et al., 2023).

### A.3.2 A SPECIAL CASE: (MINI-BATCH) SCHRÖDINGER BRIDGE CFM (TONG ET AL., 2023)

Within our general Unified Bridge Algorithm (UBA) framework (Alg. 2), we select $P_t(x|x_0, x_1)$, $Q(x_0, x_1)$ and $u_t(x|x_0, x_1)$ as follows. First in step 1,

$$P_t(x_t|x_0, x_1) = P_t^{ref}(x_t|x_0, x_1) = \mathcal{N}(x_t; (1 - t)x_0 + tx_1, \sigma_{ref}^2 t(1 - t)I) \tag{56}$$

$$Q(x_0, x_1) = P^{mEOT}(x_0, x_1) := \sum_{i \in B_0} \sum_{j \in B_1} \delta(x_0 = x_0^i)\delta(x_1 = x_1^j)p_{ij}^{mEOT} \tag{57}$$

where $(\{x_0^i\}_{i \in B_0}, \{x_1^j\}_{j \in B_1})$ is the mini-batch data, and $\{p_{ij}^{mEOT}\}_{ij}$ is a $(|B_0| \times |B_1|)$ is the mini-batch entropic OT solution matrix with the negative entropy regularizing coefficient $2\sigma_{ref}^2$ (e.g., from the Sinkhorn-Knopp algorithm) learned with the mini-batch as training data. Although the samples from the coupling distribution $Q(x_0, x_1)$ in (57) over the iterations in Alg. 2 conform to the data distributions $\pi_0$ and $\pi_1$ marginally, the mini-batch entropic OT solution is generally substantially different from the population entropic OT solution (the optimal solution of the Schrödinger Bridge).

In step 2, we choose $\sigma = 0$, and define $u_t(x_t|x_0, x_1)$ to be:

$$u_t(x_t|x_0, x_1) = \frac{1 - 2t}{2t(1 - t)}(x_t - (tx_1 + (1 - t)x_0)) + x_1 - x_0 \tag{58}$$

which can be shown to make the ODE $dx_t = u_t(x_t|x_0, x_1)dt$ admit $P_t(x_t|x_0, x_1)$ as its marginal distributions (Tong et al., 2023).

The above choices precisely yield the (mini-batch) Schrödinger Bridge CFM (SB-CFM) introduced in (Tong et al., 2023). Although the marginals of SB-CFM match those of the Schrödinger bridge solution, the entire path measure not since it only finds an ODE bridge solution.

### A.3.3 A SPECIAL CASE: DEEP SCHRÖDINGER BRIDGE MATCHING (DSBM) (SHI ET AL., 2023)

As discussed in A.2.4, the IMF/DSBM algorithm is based on the IMF principle where starting from $P(\{x_t\}_{t \in [0,1]}) = \pi_0(x_0)\pi_1(x_1)P^{ref}(\{x_t\}_{t \in (0,1)}|x_0, x_1)$, repeatedly and alternatively applying the projections $P \leftarrow \Pi_{\mathcal{M}}(P)$ and $P \leftarrow \Pi_{\mathcal{R}}(P)$ leads to convergence to the Schrödinger bridge solution. How does this algorithm fit in the framework of our Unified Bridge Algorithm (UBA) in Alg. 2? We will see that a specific choice of $P_t(x|x_0, x_1)$, $Q(x_0, x_1)$ (in step 1) and $u_t(x|x_0, x_1)$ (in step 2) precisely leads to the IMF algorithm. We describe the algorithm in Alg. 3.

In step 1, the pinned path marginals $P_t(x|x_0, x_1)$ are set to be equal to $P_t^{ref}(x|x_0, x_1)$ which can be written analytically as Gaussian (59). The coupling $Q(x_0, x_1)$ is defined to be the coupling distribution $P^{v_\theta}(x_0, x_1)$ that is induced from the SDE in the previous iteration (step 3), $P^{v_\theta} : dx_t = v_\theta(t, x_t)dt + \sigma dW_t$. In the first iteration where no $\theta$ is available yet, we set $Q(x_0, x_1) := \pi_0(x_0)\pi_1(x_1)$. We need to check if the boundary condition for (49) is satisfied. This will be done shortly in the following paragraph. In step 2, we fix $\sigma := \sigma_{ref}$, and set $u_t(x_t|x_0, x_1) := \sigma_{ref}^2 \nabla_{x_t} \log P^{ref}(x_1|x_t) = \frac{x_1 - x_t}{1 - t}$. In step 3 we update $\theta$ by solving the optimization (62), the same as (51), with the chosen $P_t(x|x_0, x_1)$, $Q(x_0, x_1)$ and $u_t(x_t|x_0, x_1)$.

Now we see how this choice leads to the IMF algorithm precisely. First, due to Doob's h-transform (Rogers & Williams, 2000), the SDE $dx_t = u_t(x_t|x_0, x_1)dt + \sigma dW_t$ with the choice (61) admits $\{P_t^{ref}(x|x_0, x_1)\}_t$ as its marginals for any $(x_0, x_1)$. Next, the step 3, if optimized perfectly and ideally with zero neural net approximation error, is equivalent to $\Pi_{\mathcal{M}}(\Pi_{\mathcal{R}}(P^{v_{\theta_{old}}}))$ where $\theta_{old}$ is the optimized $\theta$ in the previous iteration[3]. This can be easily understood by looking at the Markovian projection $\Pi_{\mathcal{M}}(P)$ written in the optimization form (48): The expectation is taken with respect to $P(x_t, x_1)$ that matches $Q(x_0, x_1)P_t^{ref}(x_t|x_0, x_1)$ in (62), and which is exactly the reciprocal projection of $P^{v_{\theta_{old}}}$ since $Q(x_0, x_1) = P^{v_{\theta_{old}}}(x_0, x_1)$ by construction. Lastly, we can verify that the choice in step 1 ensures the boundary conditions $P_0(\cdot) = \pi_0(\cdot)$, $P_1(\cdot) = \pi_1(\cdot)$. This is because $Q(x_0, x_1)$ always satisfies $Q(x_0) = \pi_0(x_0)$ and $Q(x_1) = \pi_0(x_1)$: initially $Q(x_0, x_1) = \pi_0(x_0)\pi_1(x_1)$ obviously, and later as $Q(x_0, x_1) = P^{v_\theta}(x_0, x_1)$ results from the Markovian projection (from step 3 in the previous iteration). We recall from Sec. A.2.4 that the Markovian projection preserves the boundary conditions.

• **IMF/DSBM algorithm as a UBA in minimal kinetic energy forms.** We can reformulate the IMF algorithm within our UBA framework using the minimal kinetic form as described in Alg. 4. In fact it can be shown that Alg. 3 and Alg. 4 are indeed equivalent, as stated in Theorem A.5 in Sec. A.4.2.

---

[3]Initially when there is no previous $\theta_{old}$ available, the step 3 is equivalent to $\Pi_{\mathcal{M}}(P^{init})$ where $P^{init} = \pi_0(x_0)\pi_1(x_1)P^{ref}(\cdot|x_0, x_1)$ which is already in the reciprocal class $\mathcal{R}$.

---

**Algorithm 3** IMF/DSBM algorithm (Shi et al., 2023) as a special instance of our UBA.

---

**Input:** The end-point distributions $\pi_0$ and $\pi_1$ (i.e., samples from them).
**Repeat** until convergence or a sufficient number of times:
    1. Choose a pinned marginal path $\{P_t(x|x_0, x_1)\}_t$ and a coupling distribution $Q(x_0, x_1)$ as follows:

$$P_t(x_t|x_0, x_1) := P_t^{ref}(x_t|x_0, x_1) = \mathcal{N}(x_t; (1-t)x_0 + tx_1, \sigma_{ref}^2 t(1-t)I) \tag{59}$$

$$Q(x_0, x_1) := \begin{cases} \pi_0(x_0)\pi_1(x_1) & \text{initially (if } \theta \text{ is not available)} \\ P^{v_\theta}(x_0, x_1) & \text{otherwise} \end{cases} \tag{60}$$

    2. Choose $\sigma := \sigma_{ref}$, and set $u_t(x|x_0, x_1)$ as:

$$u_t(x_t|x_0, x_1) := \sigma_{ref}^2 \nabla_{x_t} \log P^{ref}(x_1|x_t) = \frac{x_1 - x_t}{1 - t} \tag{61}$$

    3. Solve the following optimization problem with respect to the neural network $v_\theta(t, x)$:

$$\min_\theta \mathbb{E}_{t, Q(x_0, x_1)P_t(x_t|x_0, x_1)} ||u_t(x_t|x_0, x_1) - v_\theta(t, x_t)||^2 \tag{62}$$

**Return:** The learned SDE $dx_t = v_\theta(t, x_t)dt + \sigma dW_t$ as the bridge problem solution.

---

**Algorithm 4** IMF/DSBM algorithm (Shi et al., 2023) as a minimal kinetic energy form in our UBA.

---

**Input:** The end-point distributions $\pi_0$ and $\pi_1$ (i.e., samples from them).
**Repeat** until convergence or a sufficient number of times:
    1. Choose $P_t(x|x_0, x_1) = \mathcal{N}(x; \mu_t, \gamma_t^2 I)$ where $(\mu_t, \gamma_t)$ are solutions to the following optimization:

$$\arg \min_{\{\mu_t, \gamma_t\}_t} \int_0^1 \mathbb{E}_{P_t(x|x_0, x_1)} \left[ \frac{1}{2} ||\alpha_t(x|x_0, x_1)||^2 \right] dt \quad \text{where} \tag{63}$$

$$\alpha_t(x|x_0, x_1) = \frac{d\mu_t}{dt} + a_t(x - \mu_t), \quad a_t = \frac{1}{\gamma_t} \left( \frac{d\gamma_t}{dt} - \frac{\sigma_{ref}^2}{2\gamma_t} \right) \tag{64}$$

    Choose a coupling distribution $Q(x_0, x_1)$ as follows:

$$Q(x_0, x_1) := \begin{cases} \pi_0(x_0)\pi_1(x_1) & \text{initially (if } \theta \text{ is not available)} \\ P^{v_\theta}(x_0, x_1) & \text{otherwise} \end{cases} \tag{65}$$

    2. Choose $\sigma := \sigma_{ref}$, and set $u_t(x|x_0, x_1) := \alpha_t(x|x_0, x_1)$.
    3. Solve the following optimization problem with respect to the neural network $v_\theta(t, x)$:

$$\min_\theta \mathbb{E}_{t, Q(x_0, x_1)P_t(x_t|x_0, x_1)} ||u_t(x_t|x_0, x_1) - v_\theta(t, x_t)||^2 \tag{66}$$

**Return:** The learned SDE $dx_t = v_\theta(t, x_t)dt + \sigma dW_t$ as the bridge problem solution.

---

Our proof in Sec. A.4.2 relies on some results from the stochastic optimal control theory (Tzen & Raginsky, 2019). Then what is the benefit of having this stochastic optimal control formulation for the IMF algorithm? Compared to Alg 3, it has more flexibility allowing us to extend or re-purpose the bridge matching algorithm for different goals. For instance, the Generalized Schrödinger Bridge Matching (GSBM) (Liu et al., 2024) adopted a formulation similar to Alg. 4, in which they introduced the stage cost function that is minimized together with the control norm term. The final solution SDE would *not* be the Schrödinger bridge solution, but can be seen as a *generalized* solution that takes into account problem-specific stage costs. Hence the algorithmic framework in Alg. 4 is especially beneficial for developing new problem setups and novel bridge algorithms.

## A.4 THEOREMS AND PROOFS FOR UNIFIED BRIDGE ALGORITHM (UBA)

### A.4.1 PROOF OF THEOREM A.1.

To prove Theorem A.1, we show the following three lemmas in turn:

1. First, in Lemma A.2, we show that after step 2 of Alg. 2 is done, the SDE $dx_t = u_t(x_t)dt + \sigma dW_t$, $x_0 \sim \pi_0(\cdot)$ admits $\{P_t(x_t)\}_t$ as its marginal distributions where $u_t(x) = \frac{1}{P_t(x)}\mathbb{E}_Q[u_t(x|x_0, x_1)P_t(x|x_0, x_1)]$.

2. Under the assumptions made in the theorem, that is, i) the neural network $v_\theta(t, x)$'s functional space is rich enough to approximate any function arbitrarily closely; ii) the step 3 of Alg. 2 is solved perfectly, we will show that the solution to step 3 is $v_\theta(t, x) = \mathbb{E}[u_t(x|x_0, x_1)|x_t = x]$. This straightforwardly comes from the regression theorem, but we will elaborate it in greater detail in Lemma A.3 below. We then show that this conditional expectation equals $u_t(x)$ defined in (68), i.e., $v_\theta(t, x) = u_t(x)$. This will complete the proof, and we assert that $dx_t = v_{\theta^*}(t, x_t)dt + \sigma_{ref}dW_t$, $x_0 \sim \pi_0(\cdot)$ admits $\{P_t(x_t)\}_t$ as its marginals.

3. Practically, the training dynamics of the gradient descent for step 3 of Alg. 2 can be shown to be identical to that of minimizing $\mathbb{E}\|v_\theta(t, x) - u_t(x)\|^2$. This is done in Lemma A.4 although the proof is very similar to the result in (Tong et al., 2023). Hence, in practice, even without the assumptions of the ideal rich neural network functional capacity and perfect optimization, we can continue to reduce the error between $v_\theta(t, x)$ and $u_t(x)$ in the course of gradient descent for step 3.

**Lemma A.2.** *Suppose* $\{P_t(x|x_0, x_1)\}_t$ *be the marginal distributions of the SDE* $dx_t = u_t(x_t|x_0, x_1)dt + \sigma dW_t$ *for given* $x_0$ *and* $x_1$. *In other words, step 2 of Alg. 2 is done. For*

$$P_t(x) := \int P_t(x|x_0, x_1)Q(x_0, x_1)d(x_0, x_1) \tag{67}$$

$$u_t(x) := \frac{1}{P_t(x)}\mathbb{E}_{Q(x_0, x_1)}[u_t(x|x_0, x_1)P_t(x|x_0, x_1)], \tag{68}$$

*the SDE* $dx_t = u_t(x_t)dt + \sigma dW_t$, $x_0 \sim \pi_0(\cdot)$ *has marginal distributions* $\{P_t(x)\}_t$.

*Proof.* For the given $x_0$ and $x_1$, we apply the Fokker-Planck equation to the SDE $dx_t = u_t(x_t|x_0, x_1)dt + \sigma dW_t$ with its marginals $\{P_t(x|x_0, x_1)\}_t$.

$$\frac{\partial}{\partial t}P_t(x|x_0, x_1) = -\operatorname{div}\{P_t(x|x_0, x_1)u_t(x|x_0, x_1)\} + \frac{\sigma^2}{2}\Delta P_t(x|x_0, x_1) \tag{69}$$

where $\operatorname{div}$ is the divergence operator and $\Delta$ is the Laplace operator. Now we derive the Fokker-Planck equation for the target SDE as follows:

$$\frac{\partial}{\partial t}P_t(x) = \frac{\partial}{\partial t}\int P_t(x|x_0, x_1)Q(x_0, x_1)d(x_0, x_1) \tag{70}$$

$$= \int \frac{\partial}{\partial t}P_t(x|x_0, x_1)Q(x_0, x_1)d(x_0, x_1) \tag{71}$$

$$= \int \left(-\operatorname{div}\{P_t(x|x_0, x_1)u_t(x|x_0, x_1)\} + \frac{\sigma^2}{2}\Delta P_t(x|x_0, x_1)\right)Q(x_0, x_1)d(x_0, x_1) \tag{72}$$

$$= -\operatorname{div}\mathbb{E}_Q\left[u_t(x|x_0, x_1)P_t(x|x_0, x_1)\right] + \frac{\sigma^2}{2}\Delta\int P_t(x|x_0, x_1)Q(x_0, x_1)d(x_0, x_1) \tag{73}$$

$$= -\operatorname{div}\left\{P_t(x)\frac{1}{P_t(x)}\mathbb{E}_Q\left[u_t(x|x_0, x_1)P_t(x|x_0, x_1)\right]\right\} + \frac{\sigma^2}{2}\Delta P_t(x) \tag{74}$$

$$= -\operatorname{div}\{P_t(x)u_t(x)\} + \frac{\sigma^2}{2}\Delta P_t(x) \tag{75}$$

This establishes the Fokker-Planck equation for the SDE $dx_t = u_t(x_t)dt + \sigma dW_t$, $x_0 \sim \pi_0(\cdot)$, to which $\{P_t(x)\}_t$ is the solution. This completes the proof of Lemma A.2. $\qquad\square$

**Lemma A.3.** *Under the assumptions of ideal rich neural network capacity and perfect optimization made in the theorem, the solution to step 3 of Alg. 2, that is,*

$$\theta^* = \arg\min_\theta \mathbb{E}_{t, Q(x_0, x_1)P_t(x_t|x_0, x_1)}\|u_t(x_t|x_0, x_1) - v_\theta(t, x_t)\|^2 \tag{76}$$

*satisfies* $v_{\theta^*}(t, x) = \mathbb{E}[u_t(x|x_0, x_1)|x_t = x] = u_t(x)$ *where* $u_t(x)$ *is defined in (68).*

*Proof.* We prove the second equality first. The expectation $\mathbb{E}[u_t(x_t|x_0, x_1)|x_t]$ is taken with respect to the distribution $R(x_0, x_1|x_t)$ defined to be proportional to $P_t(x_t|x_0, x_1)Q(x_0, x_1)$.

$$\mathbb{E}[u_t(x_t|x_0, x_1)|x_t] = \int R(x_0, x_1|x_t)u_t(x_t|x_0, x_1)d(x_0, x_1) \tag{77}$$

$$= \int \frac{P_t(x_t|x_0, x_1)Q(x_0, x_1)}{\int P_t(x_t|x_0, x_1)Q(x_0, x_1)d(x_0, x_1)}u_t(x_t|x_0, x_1)d(x_0, x_1) \tag{78}$$

$$= \int \frac{P_t(x_t|x_0, x_1)Q(x_0, x_1)}{P_t(x_t)}u_t(x_t|x_0, x_1)d(x_0, x_1) \tag{79}$$

$$= \frac{1}{P_t(x_t)}\int P_t(x_t|x_0, x_1)Q(x_0, x_1)u_t(x_t|x_0, x_1)d(x_0, x_1) \tag{80}$$

$$= \frac{1}{P_t(x_t)}\mathbb{E}_{Q(x_0,x_1)}\big[u_t(x_t|x_0, x_1)P_t(x_t|x_0, x_1)\big] \tag{81}$$

$$= u_t(x_t) \tag{82}$$

We now prove the first equality. Although this straightforwardly comes from the regression theorem, but here we will elaborate it in greater detail. Due to the assumptions, the optimization (76) can be written in a functional form as:

$$v^* = \arg\min_{v(\cdot,\cdot)} \mathbb{E}_{t,Q(x_0,x_1)P_t(x_t|x_0,x_1)}||u_t(x_t|x_0, x_1) - v(t, x_t)||^2 \tag{83}$$

where its optimizer $v^*(t, x)$ equals the optimizer $v_{\theta^*}(t, x)$ of (76). In the functional optimization (83), the objective is completely decomposed over $t$, and we can equivalently minimize $\mathbb{E}_{Q(x_0,x_1)P_t(x_t|x_0,x_1)}||u_t(x_t|x_0, x_1) - v(t, x_t)||^2$ for each $t$. We take the functional gradient with respect to $v(t, \cdot)$. For ease of exposition, we will use simpler notation where we minimize $\mathbb{E}_{p(y,z)}||f(y, z) - g(y)||^2$ with respect to the function $g(\cdot)$. Hence there is direct correspondence: $y$ is to $x_t$, $z$ to $(x_0, x_1)$, $f(y, z)$ to $u_t(x_t|x_0, x_1)$, and $g$ to $v$. We see that at optimum,

$$\partial g(y) = \int 2p(y, z)(g(y) - f(y, z))dz = 0 \tag{84}$$

leading to $g^*(y) = \mathbb{E}[f(y, z)|y]$. This regression theorem implies $v^*(t, x_t) = \mathbb{E}[u_t(x_t|x_0, x_1)|x_t]$. This completes the proof Lemma A.3. □

**Lemma A.4.** $\nabla_\theta \mathbb{E}_{P_t(x)}||v_\theta(t, x) - u_t(x)||^2 = \nabla_\theta \mathbb{E}_{P_t(x|x_0,x_1)Q(x_0,x_1)}||v_\theta(t, x) - u_t(x|x_0, x_1)||^2$ *for each $t$, where $P_t(x)$ and $u_t(x)$ are defined as (67) and (68), respectively.*

*Proof.*

$$\nabla_\theta \mathbb{E}_{P_t(x)}||v_\theta(t, x) - u_t(x)||^2 = \nabla_\theta \mathbb{E}_{P_t(x)}\Big[||v_\theta(t, x)||^2 - 2v_\theta(t, x)^\top u_t(x)\Big] \tag{85}$$

$$= \nabla_\theta \mathbb{E}_{P_t(x)}||v_\theta(t, x)||^2 - 2\nabla_\theta \mathbb{E}_{P_t(x)}\Big[v_\theta(t, x)^\top u_t(x)\Big] \tag{86}$$

The first term can be written as:

$$\nabla_\theta \mathbb{E}_{P_t(x)}||v_\theta(t, x)||^2 = \nabla_\theta \int ||v_\theta(t, x)||^2 P_t(x)dx \tag{87}$$

$$= \nabla_\theta \int ||v_\theta(t, x)||^2 P_t(x|x_0, x_1)Q(x_0, x_1)d(x, x_0, x_1) \tag{88}$$

$$= \nabla_\theta \mathbb{E}_{P_t(x|x_0,x_1)Q(x_0,x_1)}||v_\theta(t, x)||^2 \tag{89}$$

The second term can be derived as:

$$\nabla_\theta \mathbb{E}_{P_t(x)}\Big[v_\theta(t, x)^\top u_t(x)\Big] = \nabla_\theta \int v_\theta(t, x)^\top u_t(x)P_t(x)dx \tag{90}$$

$$= \nabla_\theta \int v_\theta(t, x)^\top \mathbb{E}_{Q(x_0,x_1)}[u_t(x|x_0, x_1)P_t(x|x_0, x_1)]dx \tag{91}$$

$$= \nabla_\theta \int v_\theta(t, x)^\top u_t(x|x_0, x_1)P_t(x|x_0, x_1)Q(x_0, x_1)d(x, x_0, x_1) \tag{92}$$

$$= \nabla_\theta \mathbb{E}_{P_t(x|x_0,x_1)Q(x_0,x_1)}\Big[v_\theta(t, x)^\top u_t(x|x_0, x_1)\Big] \tag{93}$$

Combining (89) and (93), and noting that $||u_t(x|x_0, x_1)||^2$ is independent of $\theta$, we complete the proof of Lemma A.4. $\qquad\square$

### A.4.2 EQUIVALENCE BETWEEN ALG. 3 AND ALG. 4

**Theorem A.5.** *Under the same assumptions as those in Theorem A.1, Alg. 3 and Alg. 4 lead to the same SDE solution, which is the Schrödinger bridge matching.*

*Proof.* The proof goes as follows: i) We first show that the optimization problem (63–64) in step 1 in Alg. 4 can be seen as a constrained minimal kinetic energy optimal control problem with the constraint of Gaussian marginals $\{P_t(x|x_0, x_1)\}_t$; ii) We then relax it to an unconstrained version, and view the unconstrained problem as an instance of stochastic optimal control problem with fixed initial; iii) The latter is then shown to admit Gaussian pinned marginal solutions following the theory developed in (Tzen & Raginsky, 2019), thus proving that Gaussian constraining does not essentially restrict the problem. It turns out that the optimal pinned marginals and the optimal control have exactly the linear interpolation forms in (59) and (61), respectively, which completes the proof.

In step 2, since we set $u_t(x|x_0, x_1) = \alpha_t(x|x_0, x_1)$ where $\alpha$ is the solution to (63–64), we will use the notation $u$ in place of $\alpha$ throughout the proof. First, we show that the SDE $dx_t = u_t(x_t|x_0, x_1)dt + \sigma dW_t$ with initial state $x_0$ at $t=0$ and $u$ satisfying (64), admits $\{P_t(x_t|x_0, x_1) = \mathcal{N}(x_t; \mu_t, \gamma_t^2 I)\}_t$ as marginal distributions. Note that we must have $\mu_0 = x_0$, $\mu_1 = x_1$, $\gamma_0 \to 0$, and $\gamma_1 \to 0$ due to the conditioning (pinned process). This fact is in fact an extension of the similar one for ODE cases in (Tong et al., 2023). We will do the proof here for SDE cases. We will establish the Fokker-Planck equation for the SDE, and we derive:

$$\frac{\partial P_t(x|x_0, x_1)}{\partial t} = P_t(x|x_0, x_1) \cdot \frac{\partial \log \mathcal{N}(x; \mu_t, \gamma_t^2 I)}{\partial t} \tag{94}$$

$$= P_t(x|x_0, x_1) \cdot \left( -\frac{\gamma_t'}{\gamma_t}d + \frac{(x - \mu_t)^\top \mu_t'}{\gamma_t^2} + \frac{||x - \mu_t||^2 \gamma_t'}{\gamma_t^3} \right) \tag{95}$$

where $\mu_t'$ and $\gamma_t'$ are the time derivatives. We also derive the divergence and Laplacian as follows:

$$\text{div}\{P_t(x|x_0, x_1)u_t(x|x_0, x_1)\} =$$

$$- P_t(x|x_0, x_1) \cdot \left( -\frac{\gamma_t'}{\gamma_t}d + \frac{(x - \mu_t)^\top \mu_t'}{\gamma_t^2} + \frac{||x - \mu_t||^2 \left(\gamma_t' - \frac{\sigma^2}{2\gamma_t}\right)}{\gamma_t^3} + \frac{\sigma^2}{2\gamma_t^2}d \right) \tag{96}$$

$$\Delta P_t(x|x_0, x_1) = \text{Tr}\left(\nabla_x^2 P_t(x|x_0, x_1)\right) = P_t(x|x_0, x_1) \cdot \left( \frac{||x - \mu_t||^2}{\gamma_t^4} - \frac{d}{\gamma_t^2} \right) \tag{97}$$

From (95), (96) and (97), we can establish the following equality, and it proves the fact.

$$\frac{\partial P_t(x|x_0, x_1)}{\partial t} = -\text{div}\{P_t(x|x_0, x_1)u_t(x|x_0, x_1)\} + \frac{\sigma^2}{2}\Delta P_t(x|x_0, x_1) \tag{98}$$

From the above fact, we can re-state the step 1 of Alg. 4 as follows:

*(Step 1 re-stated) Choose $P_t(x|x_0, x_1)$ as the marginals of the SDE, $dx_t = u_t(x_t|x_0, x_1)dt + \sigma dW_t$ with initial state $x_0$ at $t=0$ where $u$ is the solution to the constrained optimization:*

$$\min_u \int_0^1 \mathbb{E}_{P_t(x|x_0, x_1)}\left[\frac{1}{2}||u_t(x|x_0, x_1)||^2\right] dt \ \text{ s.t. } \ \{P_t(x|x_0, x_1)\}_t \text{ are Gaussians} \tag{99}$$

Instead of solving (99) directly, we try to deal with its unconstrained version, i.e., without the Gaussian marginal constraint. To this end we utilize the theory of stochastic optimal control with the fixed initial state (Tzen & Raginsky, 2019), which we adapted for our purpose below in Lemma A.6.

In Lemma A.6, we adopt the Dirac's delta function $g(\cdot) = \delta_{x_1}$ for the terminal cost to ensure that the SDE $dx_t = u_t(x_t|x_0, x_1)dt + \sigma dW_t$ with initial state $x_0$ lands at $x_1$ as the final state. Then the unconstrained version of (99), which is perfectly framed as an optimal control problem in Lemma A.6, has the optimal solution written as:

$$u_t(x|x_0, x_1) = \sigma^2 \nabla_x \log \mathbb{E}_{P^{ref}}[\delta_{x_1}|x_t = x] = \sigma^2 \nabla_x \log P^{ref}(x_1|x_t = x) \tag{100}$$

which coincides with (61) in Alg. 3. Now, due to Doob's h-transform (Rogers & Williams, 2000), the SDE $dx_t = u_t(x_t|x_0, x_1)dt + \sigma dW_t$ with the choice (61) or (100) admits $\{P_t^{ref}(x|x_0, x_1)\}_t$ as its marginals. In other words, $P_t(x|x_0, x_1) = P_t^{ref}(x|x_0, x_1)$, which is Gaussian, meaning that the constrained optimization (99) and its unconstrained version essentially solve the same problem.

Noting that (59) and (61) in Alg. 3 are equivalent to (63) and (64) in Alg. 4, we conclude that Alg. 3 and Alg. 4 are equivalent. $\qquad\square$

Now we describe Lemma A.6.

**Lemma A.6** (Stochastic optimal control with fixed initial state; Adapted from Theorem 2.1 in (Tzen & Raginsky, 2019)). *Let $P^b$ be the path measure of the SDE: $dx_t = b_t(x)dt + \sigma dW_t$, starting from the fixed initial state $x_0$. For the stochastic optimal control problem with the immediate cost $\frac{1}{2\sigma^2}||b_t(x_t)||^2$ at time $t$ and the terminal cost $\log 1/g(x_1)$ at final time $t = 1$ for any function $g$, the cost-to-go function defined as:*

$$J_t^b(x) := \mathbb{E}_{P^b}\left[\int_t^1 \frac{1}{2\sigma^2}||b_t(x_t)||^2 - \log g(x_1)\Big|x_t = x\right] \tag{101}$$

*has the optimal control (i.e., the optimal drift $b_t(x)$)*

$$b_t^*(x) = \arg\min_b J_t^b(x) = \sigma^2 \nabla_x \log \mathbb{E}_{P^{ref}}[g(x_1)|x_t = x] \tag{102}$$

*where $P^{ref}$ is the Brownian path measure with diffusion coefficient $\sigma$.*

*Proof.* We utilize the (simplified) Feynman–Kac formula, saying that the PDE,

$$\frac{\partial h_t(x)}{\partial t} + \mu_t(x)^\top \nabla_x h_t(x) + \frac{1}{2}\text{Tr}\Big(\sigma^2 \nabla_x^2 h_t(x)\Big) = 0, \quad h_1(\cdot) = q(\cdot) \tag{103}$$

has a solution $h_t(x) = \mathbb{E}[q(x_1)|x_t = x]$ where the expectation is taken with respect to the SDE, $dx_t = \mu_t(x_t)dt + \sigma dW_t$.

Now we plug in $\mu_t = 0$, $q = g$, and let $v_t(x) := -\log h_t(x)$. Note that $h_t(x)$ is always positive since $g$ is positive, and hence $v_t(x)$ is well defined. Then by some algebra, we see the following PDE:

$$\frac{\partial v_t(x)}{\partial t} + \frac{1}{2}\text{Tr}\Big(\sigma^2 \nabla_x^2 v_t(x)\Big) = \frac{\sigma^2}{2}||\nabla_x v_t(x)||^2, \quad v_1(\cdot) = -\log g(\cdot) \tag{104}$$

has a solution $v_t(x) = -\log \mathbb{E}_{P^{ref}}[g(x_1)|x_t = x]$. Note that we can write $\frac{\sigma^2}{2}||\nabla_x v_t(x)||^2$ as the following variational form,

$$\frac{\sigma^2}{2}||\nabla_x v_t(x)||^2 = -\min_b \; b^\top \nabla_x v_t(x) + \frac{||b||^2}{2\sigma^2} \tag{105}$$

where the minimum is attained at $b^* = -\sigma^2 \nabla_x v_t(x)$. So $v_t(x) = -\log \mathbb{E}_{P^{ref}}[g(x_1)|x_t = x]$ is the solution to:

$$\frac{\partial v_t(x)}{\partial t} + \frac{1}{2}\text{Tr}\Big(\sigma^2 \nabla_x^2 v_t(x)\Big) = -\min_b \; b^\top \nabla_x v_t(x) + \frac{||b||^2}{2\sigma^2}, \quad v_1(\cdot) = -\log g(\cdot) \tag{106}$$

Note that (106) is the Hamilton-Jacobi-Bellman equation for the stochastic optimal control problem with the immediate cost $\frac{1}{2\sigma^2}||b_t(x_t)||^2$ and the terminal cost $\log 1/g(x_1)$. In fact, $v_t(x) = -\log \mathbb{E}_{P^{ref}}[g(x_1)|x_t = x]$ is the (optimal) value function, and the optimal control, i.e., the solution to (105) in a function form, is: $b_t^*(x) = -\sigma^2 \nabla_x v_t(x) = \sigma^2 \nabla_x \log \mathbb{E}_{P^{ref}}[g(x_1)|x_t = x]$. $\qquad\square$

# B  TECHNICAL DETAILS OF GAUSSIAN PROCESS GSBM

## B.1  ELBO DERIVATIONS

First, the posterior path measure can be written as:

$$\mathcal{P}_{post}(P_\bullet) = \frac{\mathcal{P}_{prior}(P_\bullet) \cdot \exp(-J(P_\bullet; V_\bullet)/\tau)}{E(\eta, \tau)} \tag{107}$$

where $E(\eta, \tau)$ is the evidence that depends only on the model parameters $\eta$ (prior kernel hyperparameters) and $\tau$ (the likelihood hyperparameter), and defined as a normalizer:

$$E(\eta, \tau) = \mathbb{E}_{\mathcal{P}_{prior}(P_\bullet)}\big[\exp(-J(P_\bullet; V_\bullet)/\tau)\big] \tag{108}$$

We start with the KL divergence between the variational posterior and the true posterior measures.

$$\mathrm{KL}(\mathcal{Q}(P_\bullet)||\mathcal{P}_{post}(P_\bullet)) = \mathbb{E}_{\mathcal{Q}(P_\bullet)}\left[\log \frac{d\mathcal{Q}(P_\bullet)}{d\mathcal{P}_{post}(P_\bullet)}\right] \tag{109}$$

$$= \log E(\eta, \tau) + \mathbb{E}_{\mathcal{Q}(P_\bullet)}\big[J(P_\bullet; V_\bullet)/\tau\big] + \mathrm{KL}(\mathcal{Q}(P_\bullet)||\mathcal{P}_{prior}(P_\bullet)) \tag{110}$$

Now let $Y$ and $Y'$ be arbitrary time index sets in $[0, 1]$. The last term in (110) can be written as:

$$\mathrm{KL}(\mathcal{Q}(\mu_Y, \mu_Z, \tilde{\gamma}_{Y'}, \tilde{\gamma}_{Z'})||\mathcal{P}_{prior}(\mu_Y, \mu_Z, \tilde{\gamma}_{Y'}, \tilde{\gamma}_{Z'})) = \tag{111}$$

$$= \mathbb{E}_\mathcal{Q}\left[\log \frac{\mathcal{Q}(\mu_Z) \cdot \cancel{\mathcal{P}_{prior}(\mu_Y|\mu_Z)} \cdot \mathcal{Q}(\tilde{\gamma}_{Z'}) \cdot \cancel{\mathcal{P}_{prior}(\tilde{\gamma}_{Y'}|\tilde{\gamma}_{Z'})}}{\mathcal{P}_{prior}(\mu_Z) \cdot \cancel{\mathcal{P}_{prior}(\mu_Y|\mu_Z)} \cdot \mathcal{P}_{prior}(\tilde{\gamma}_{Z'}) \cdot \cancel{\mathcal{P}_{prior}(\tilde{\gamma}_{Y'}|\tilde{\gamma}_{Z'})}}\right] \tag{112}$$

$$= \mathbb{E}_\mathcal{Q}\left[\log \frac{\mathcal{Q}(\mu_Z) \cdot \mathcal{Q}(\tilde{\gamma}_{Z'})}{\mathcal{P}_{prior}(\mu_Z) \cdot \mathcal{P}_{prior}(\tilde{\gamma}_{Z'})}\right] \tag{113}$$

$$= \mathrm{KL}(\mathcal{Q}(\mu_Z)||\mathcal{P}_{prior}(\mu_Z)) + \mathrm{KL}(\mathcal{Q}(\tilde{\gamma}_{Z'})||\mathcal{P}_{prior}(\tilde{\gamma}_{Z'})) \tag{114}$$

$$= \sum_{j=1}^d \mathrm{KL}\big(\mathcal{N}(C^{\mu,j}, S^{\mu,j})||\mathcal{N}(M_Z^{\mu,j}, L_{Z,Z}^{\mu,j})\big) + \mathrm{KL}\big(\mathcal{N}(C^{\tilde{\gamma}}, S^{\tilde{\gamma}})||\mathcal{N}(m_{Z'}^{\tilde{\gamma}}, k_{Z',Z'}^{\tilde{\gamma}})\big) \tag{115}$$

where (112) comes from our construction of the variational density in (14) with $\mathcal{Q}(\mu_\bullet|\mu_Z)$ set equal to $\mathcal{P}_{prior}(\mu_\bullet|\mu_Z)$ (and similarly for $\tilde{\gamma}$). Eq. (115) is due to our Gaussian modeling of the inducing variables in (15). Note that this derivation starting from index sets $Y$ and $Y'$ results in the quantity (115) that does not depend on them. That is, (115) can be safely plugged in for the last term of (110), which we denote by $\mathrm{KL}(\mathcal{Q}(P_{Z,Z'})||\mathcal{P}_{prior}(P_{Z,Z'}))$. Due to the non-negativity of $\mathrm{KL}(\mathcal{Q}(P_\bullet)||\mathcal{P}_{post}(P_\bullet))$, we have:

$$\log E(\eta, \tau) \geq -\Big(\mathbb{E}_{\mathcal{Q}(P_\bullet)}\big[J(P_\bullet; V_\bullet)/\tau\big] + \mathrm{KL}(\mathcal{Q}(P_{Z,Z'})||\mathcal{P}_{prior}(P_{Z,Z'}))\Big) \tag{116}$$

where the right hand side is the evidence lower bound objective (ELBO).

Lastly, the variational-prior KL divergence $\mathrm{KL}(\mathcal{Q}(P_{Z,Z'})||\mathcal{P}_{prior}(P_{Z,Z'}))$, is confined to $n$-dim Gaussians, and can be written in closed forms as follows:

$$\mathrm{KL}(\mathcal{Q}(P_{Z,Z'})||\mathcal{P}_{prior}(P_{Z,Z'})) = \tag{117}$$

$$\frac{d}{2}\left(\mathrm{Tr}\Big((L_{Z,Z}^\mu)^{-1}\big(\overline{S}^\mu + \overline{R}^\mu\big)\Big) - n + \log|L_{Z,Z}^\mu| - \overline{G}^\mu\right) + \tag{118}$$

$$\frac{1}{2}\left(\mathrm{Tr}\Big((k_{Z',Z'}^{\tilde{\gamma}})^{-1}\big(S^{\tilde{\gamma}} + (C^{\tilde{\gamma}} - m_{Z'}^{\tilde{\gamma}})(C^{\tilde{\gamma}} - m_{Z'}^{\tilde{\gamma}})^\top\big)\Big) - n' + \log|k_{Z',Z'}^{\tilde{\gamma}}| - \log|S^{\tilde{\gamma}}|\right) \tag{119}$$

where we assumed the same kernel functions over all dimensions $j$ of $\mu_t$, denoted by $L_{Z,Z}^\mu := L_{Z,Z}^{\mu,j}$ $\forall j$, $n' := |Z'|$, and defined as:

$$\overline{S}^\mu = \frac{1}{d}\sum_{j=1}^d S^{\mu,j}, \quad \overline{R}^\mu = \frac{1}{d}\sum_{j=1}^d (C^{\mu,j} - M_Z^{\mu,j})(C^{\mu,j} - M_Z^{\mu,j})^\top, \quad \overline{G}^\mu = \frac{1}{d}\sum_{j=1}^d \log|S^{\mu,j}| \tag{120}$$

## B.2 Complexity Analysis

In this section we analyze the computational complexity of our GP-GSBM and GSBM. Since the SDE training part in both methods are identical, we mainly compare the ELBO learning of our GP-GSBM with the corresponding CondSOC optimization in GSBM. As described in Sec. 3, the key steps in ELBO learning consist of: i) sampling $(\mu_t, \gamma_t)$ from the variational posterior $\mathcal{Q}$, ii) computing $\alpha_t(x_t|x_0, x_1)$ which involves time derivatives (e.g., $\dot{\mu}_t$), iii) evaluate the likelihood $J(P_\bullet; V_\bullet)$, and

Table 5: Complexity analysis for ELBO learning in our GP-GSBM and CondSOC optimization in GSBM (Liu et al., 2024).

|  | GSBM (LIU ET AL., 2024) | GP-GSBM (OURS) |
|---|---|---|
| SAMPLE/ESTIMATE $(\mu_t, \gamma_t)$ | $O(NTnd)$ | $O(NTnd + \tilde{N}Tn^3)$ |
| COMPUTE $\alpha_t(x_t|x_0, x_1)$ | $O(Td)$ | $O(Td)$ |
| EVALUATE LIKELIHOOD/LOSS | $O(T \cdot (d + T_V))$ | $O(T \cdot (d + T_V))$ |
| EVALUATE KL$(\mathcal{Q}(P_Z)||\mathcal{P}_{prior}(P_Z))$ | $-$ | $O(\tilde{N}n^3 + Nn^2d)$ |

iv) evaluate the variational-prior KL term. The former three steps decide the first term of (21) while the latter determines the second term. The first three steps have corresponding counterparts in GSBM: i) estimating $(\mu_t, \gamma_t)$ from the spline model, ii) computing $\alpha_t(x_t|x_0, x_1)$ , and iii) evaluating the loss in (3). It does not have the KL computation.

First, the $(\mu_t, \gamma_t)$ sampling step in our GP-GSBM. Following (17–20) and recalling that we used diagonal $S$ and the same kernel function across all dimensions $j$, sampling $(\mu_t, \gamma_t)$ takes $O(NTnd + \tilde{N}Tn^3)$ time where $T$ is the batch size, $N$ is the number of ELBO learning iterations, and $\tilde{N} = 1$ if no model selection for the kernel hyperparameters $\eta$ is required, and $\tilde{N} = N$ otherwise. This is because if the kernel function is fixed, the kernel matrix inversion can be amortized over iterations by caching inverse kernel matrices. Using the same notation in GSBM, where $n$ denotes the number of spline knot points, the spline estimation of $(\mu_t, \gamma_t)$, corresponding to the sampling in our case, takes $O(NTnd)$.

Computing $\alpha_t(x_t|x_0, x_1)$ in (5) takes $O(Td)$ for both our approximate time derivatives and GSBM's auto-differentiation. Evaluating $J(P_\bullet; V_\bullet)$ in (7) takes $O(T \cdot (d + T_V))$ where $T_V$ is the cost of computing $V_t(x_t)$. GSBM incurs the same cost for the loss evaluation. The variational-prior KL term, only appearing in our GP-GSBM, takes $O(\tilde{N}n^3 + Nn^2d)$ complexity following (117–120).

We summarize these results in Table 5. Comparing with GSBM, we have additional cost, mainly $O(n^3)$ kernel inversion cost when model selection is performed for the kernel hyperparameters $\eta$ and $O(n^2d)$ in the KL divergence computation. This may be costly for large $d$ even though we have constant $n$ (usually no greater than 30). One work around is to use diagonalized kernels in (118–119), which reduces the complexity to linear in $n$. In practice, we observed no significant difference from using full kernel matrices. In this case, GP-GSBM incurs the same asymptotic complexity as GSBM when we do not use model selection during ELBO learning.

## C  EXPERIMENTAL DETAILS

We mostly follow the experimental settings, default hyperparameters, and network architectures from GSBM (Liu et al., 2024) and their official code base[4]. DSBM (Shi et al., 2023) is performed by turning off the CondSOC optimization in GSBM, which initializes the marginal path as linear interpolation of DSBM. Our GP-GSBM is also implemented based on their code base where we similarly take alternating forward and backward SDE training and integration for coupling $Q(x_0, x_1)$ samples. For the kernel implementation, we adopted the GPyTorch[5] library (Gardner et al., 2018).

**Problem setups.** For the stage cost for crowd navigation in (24), we use $\lambda_{obs} = 1500, \lambda_{ent} = 0, \lambda_{cgst} = 50$ for Stunnel, and $\lambda_{obs} = 1500, \lambda_{ent} = 5, \lambda_{cgst} = 0$ for GMM. For the noisy scenario in LiDAR, the manifold projection is altered by $(I + 0.5\epsilon)$ in a multiplicative way where $\epsilon \sim \mathcal{N}(0, I)$.

**Hyperparameters.** The numbers of inducing points in our GP-GSBM, $n$ for $\mu_t$ and $n'$ for $\gamma_t$, correspond to the numbers of spline knot points in GSBM. And we match them for both models for fair comparison. Specifically we chose: Stunnel and GMM ($n = 15$, $n' = 30$), LiDAR ($n = 30$, $n' = 30$), and CIFAR-10 and AFHQ ($n = 8$, $n' = 8$). For all experiments, our variational inference ELBO learning, corresponding to the CondSOC optimization in GSBM, takes the number of iterations: 1000 (Stunnel), 2000 (GMM), 200 (LiDAR), and 100 (CIFAR-10 and AFHQ). Following (Liu et al., 2024), we use the Adam optimizer (Kingma & Ba, 2015) for AFHQ (and CIFAR-10), SGD with momentum for LiDAR, and SGD for the rest. For the SDE noise level, $\sigma = 1.0$ is used for crowd

---

[4]https://github.com/facebookresearch/generalized-schrodinger-bridge-matching
[5]https://github.com/cornellius-gp/gpytorch

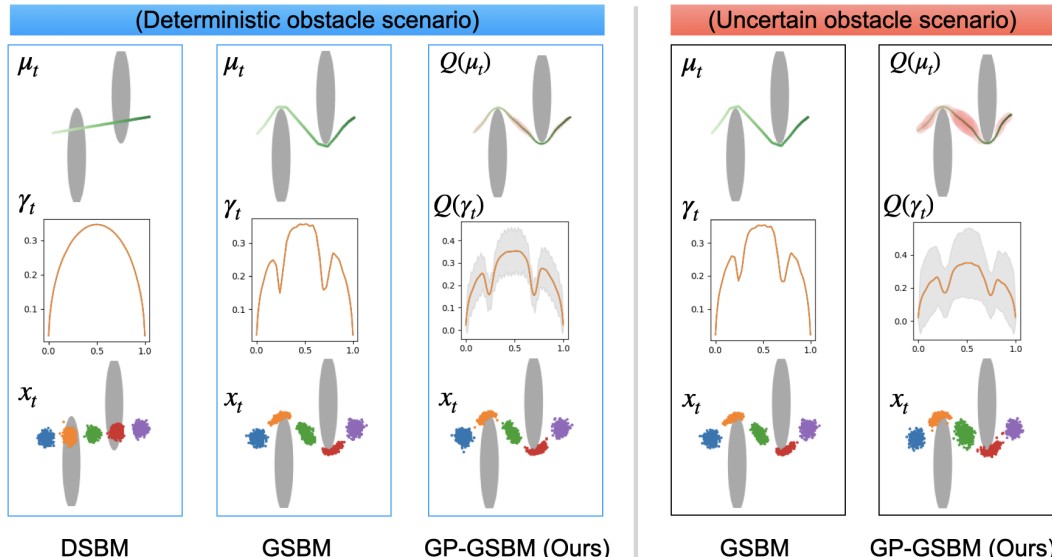

Figure 4: Stunnel results. The two ellipsoidal obstacles are shown as gray regions. For each panel, we show $\mu_t$, $\gamma_t$ (top and middle rows) of the learned pinned marginals $P_t(x_t|x_0, x_1) = \mathcal{N}(\mu_t, \gamma_t^2 I)$, and samples from the pinned marginal at 5 uniform time points (bottom row). The two boundary distributions are isotropic Gaussians where $\pi_0$ is shown as blue points, and $\pi_1$ as purple. For DSBM (Shi et al., 2023) and GSBM (Liu et al., 2024), the learned $\mu_t$, $\gamma_t$ are shown. For our GP-GSBM the learned variational posteriors $\mathcal{Q}(\mu_t)$ and $\mathcal{Q}(\gamma_t)$ are depicted as means and $2\times$ pointwise stdevs (95% confidence intervals), the latter as red-shaded regions.

navigation tasks, and $\sigma = 0.5$ for AFHQ. For CIFAR-10 (a non-GSB task), small $\sigma$ is known to be preferred as alluded in (Shi et al., 2023), and we use $\sigma = 0.001$. For SDE integration, we take 1000 time steps. For any other hyperparameters that are not discussed here, we take the default values in the official code base of (Liu et al., 2024).

**Model selection.** Due to the extra cost of the gradient-based kernel hyperparameter ($\eta$) search, we use a grid-based empirical Bayes. More specifically, for the squared exponential kernel that we used in our experiments, the width parameter $\eta$ is chosen from the candidate set $\{0.05, 0.1, 0.5, 1.0\}$, where we select the one that yields the highest ELBO score. This allows for expedited GP inference since the kernel inversion can be done only once at the beginning and amortized over variational inference iterations. For the likelihood hyperparameter $\tau$ we use gradient-based empirical Bayes.

**Network architectures.** We describe the architectures for the SDE network $v_\theta(t, x)$. For the crowd navigation problems, we adopt the fully connected network, the same as (Liu et al., 2022), that has 4 to 5 residual blocks with sinusoidal time embeddings. For the AFHQ task, we use the U-Net (Ronneberger et al., 2015) architecture with implementation from (Dhariwal & Nichol, 2021). For the CIFAR-10 experiment, we follow the same U-Net architecture as that implemented in CFM (Tong et al., 2023), which uses 4 heads and 16 attention resolutions with two residual blocks. All networks except for the CIFAR-10 case are trained from scratch, and optimized with AdamW (Loshchilov & Hutter, 2019). For CIFAR-10, we warm-started training with the trained CFM checkpoint similarly as (Shi et al., 2023). For inference, we use the exponential moving average with rate 0.9999.

**Computing platforms.** All experiments for Stunnel and GMM were conducted on a single RTX-2080-Ti GPU while two RTX-2080-Ti GPUs were used for CIFAR-10. For LiDAR, a single RTX-4090 GPU was used. For AFHQ, we train our GP-GSBM on 32 V100 GPUs (4 nodes and 8 GPUs per node) where a similar setup was used in GSBM training in (Liu et al., 2024).

## D ADDITIONAL EXPERIMENTAL RESULTS

### D.1 MORE FIGURES FOR AFHQ IMAGE-TO-IMAGE TRANSLATION (DOG → CAT)

After training the model, we pair the coupling of $(x_1, x_0)$ by running the trained (reverse-time) SDE model starting from $x_1 \sim \pi_1$, which is shown in Fig. 6, where $x_1$ (data) samples are on the rightmost

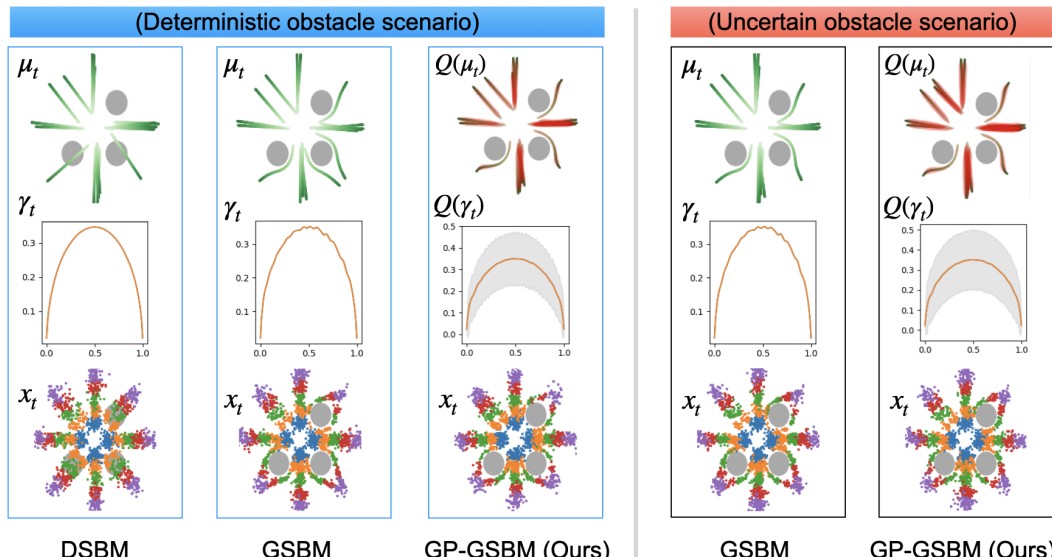

Figure 5: GMM results. The three blob obstacles are shown as gray regions. For each panel, we show $\mu_t$, $\gamma_t$ (top and middle rows) of the learned pinned marginals $P_t(x_t|x_0, x_1) = \mathcal{N}(\mu_t, \gamma_t^2 I)$, and samples from the pinned marginal at 5 uniform time points (bottom row). The two boundary distributions are mixtures of isotropic Gaussians where $\pi_0$ is shown as blue points (4 Gaussian blobs in the center), and $\pi_1$ as purple (8 outer Gaussian blobs). For DSBM (Shi et al., 2023) and GSBM (Liu et al., 2024), the learned $\mu_t$, $\gamma_t$ are shown. For our GP-GSBM, the learned variational posteriors $\mathcal{Q}(\mu_t)$ and $\mathcal{Q}(\gamma_t)$ are depicted as means and $2\times$ pointwise stdevs (95% confidence intervals), the latter as red-shaded regions.

Table 6: (CIFAR-10) FID scores on this non-GSB image generation problem. The results from CFM and Stream-level GP are excerpted from (Wei & Ma, 2025).

| CFM | | STREAM-LEVEL GP | | GP-GSBM (OURS) |
|---|---|---|---|---|
| I-CFM | OT-CFM | I-CFM | OT-CFM | |
| $3.75 \pm 0.006$ | $3.74 \pm 0.009$ | $3.62 \pm 0.008$ | $3.75 \pm 0.009$ | $3.65 \pm 0.008$ |

columns, and $x_0$ (generated) samples on the leftmost. From this coupling, we show the mean images from $P_t(x|x_0, x_1)$ for the prior and posterior (before and after the variational GP inference) in the middle. Visually clearly, the posterior exhibits semantically more meaningful path samples than the prior with less artifacts since the posterior takes into account the latent-preserving stage cost, forcing the samples to stay closer to the image manifold. In Fig. 7, the SDE generation progress is shown with intermediate time steps. We also visualize the generated cat images in Fig. 8.

### D.2 CIFAR-10 RESULTS (NON-GSB COMPARISON TO STREAM-LEVEL GP)

We follow the typical CIFAR-10 experimental setup from (Tong et al., 2023; Wei & Ma, 2025) where $\pi_0 = \mathcal{N}(0, I)$ and $\pi_1$ is the CIFAR-10 data. First we note that this experimental setting may not be an ideal example for SB matching problem as indicated by the fact that CFM with independent coupling (I-CFM) is comparable or sometimes better than mini-batch OT-based coupling (OT-CFM).

For the FID computation, we use $50K$ samples generated and averaged over 20 random runs. The results are summarized in Table 6 where means and standard errors are reported. Comparing our GP-GSBM, stream-level GP, and CFM, most methods perform equally well on this (non-GSB) noise-to-image generation problem. Some images generated by our model are shown in Fig. 9 (diffusion from $\pi_0$ to $\pi_1$) and Fig. 10 (final generated samples).

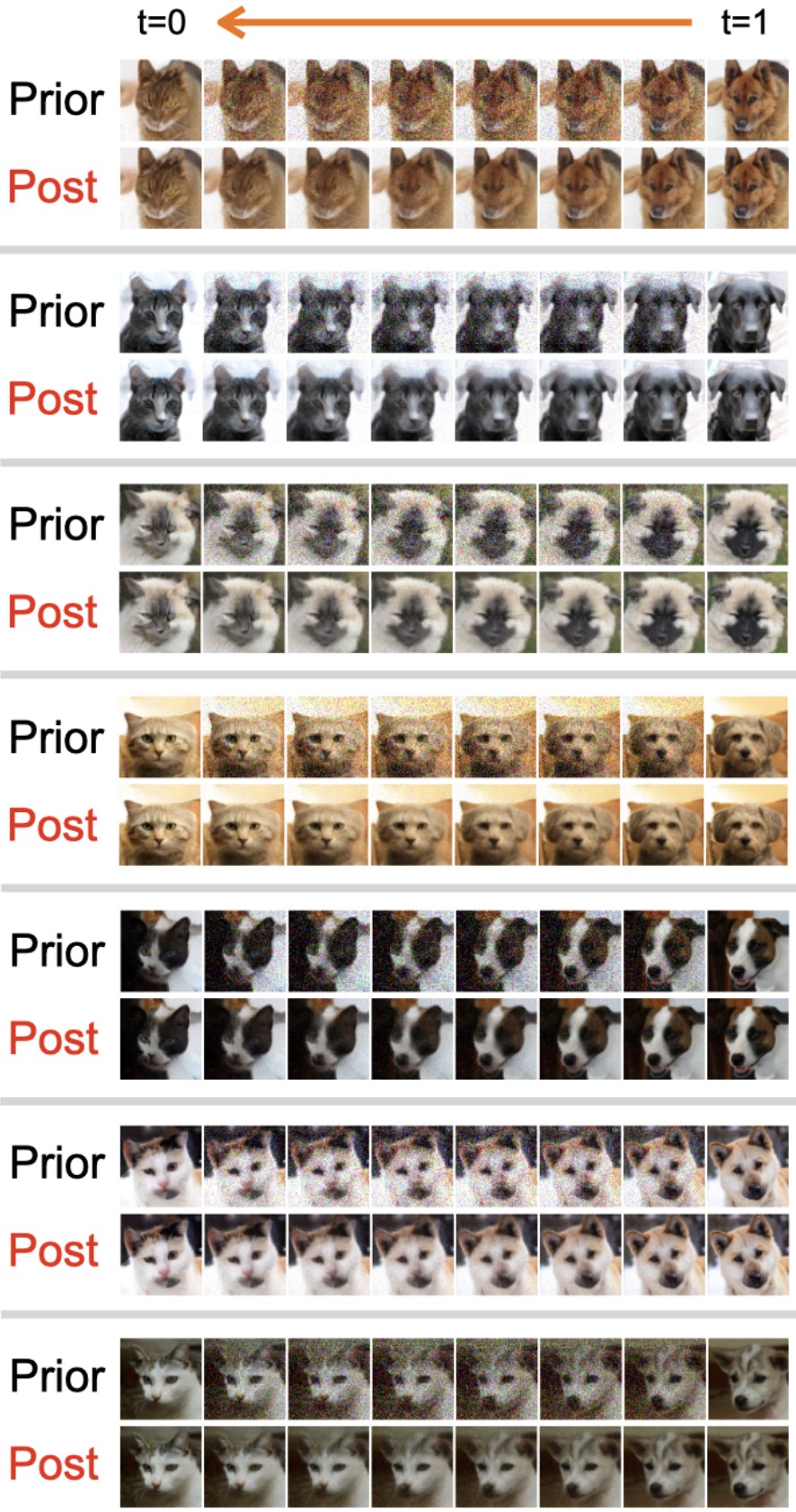

Figure 6: GP-GSBM on AFHQ dog ($t\!=\!1$, rightmost columns) $\rightarrow$ cat ($t\!=\!0$, leftmost columns). Mean images from $P_t(x|x_0, x_1)$ for the prior and posterior (before and after the variational GP inference).

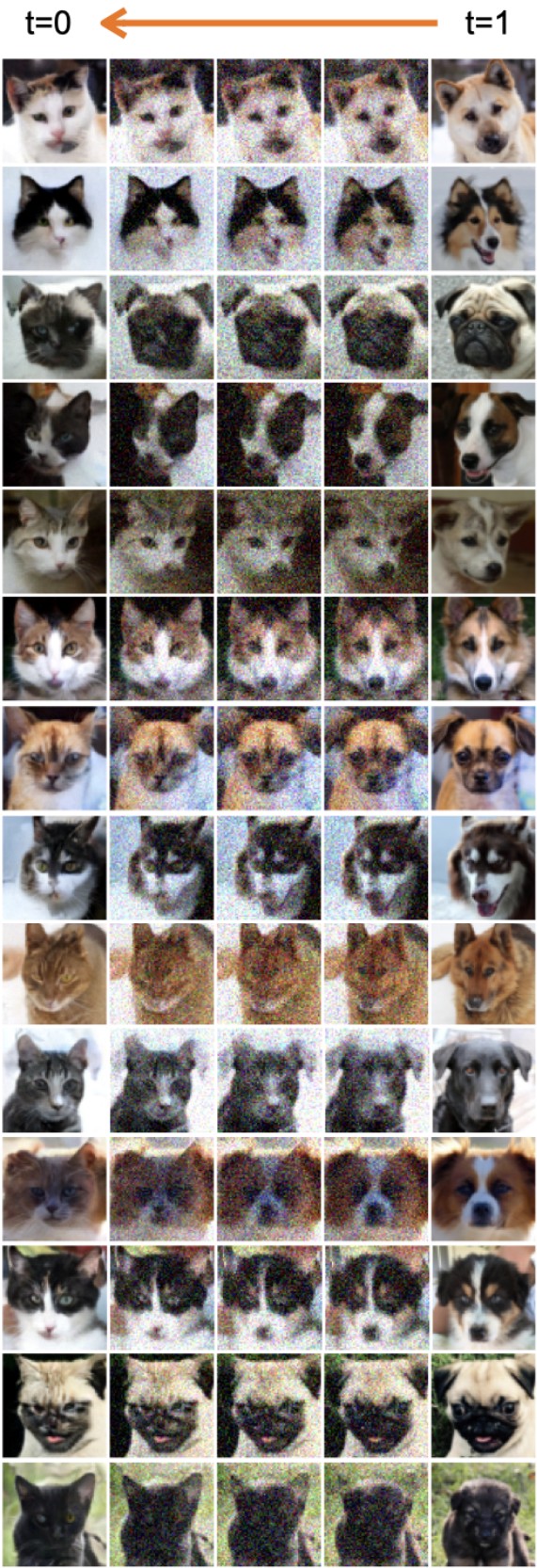

Figure 7: GP-GSBM on AFHQ dog ($t=1$, rightmost columns) → cat ($t=0$, leftmost columns). The (reverse-time) SDE generation progress.

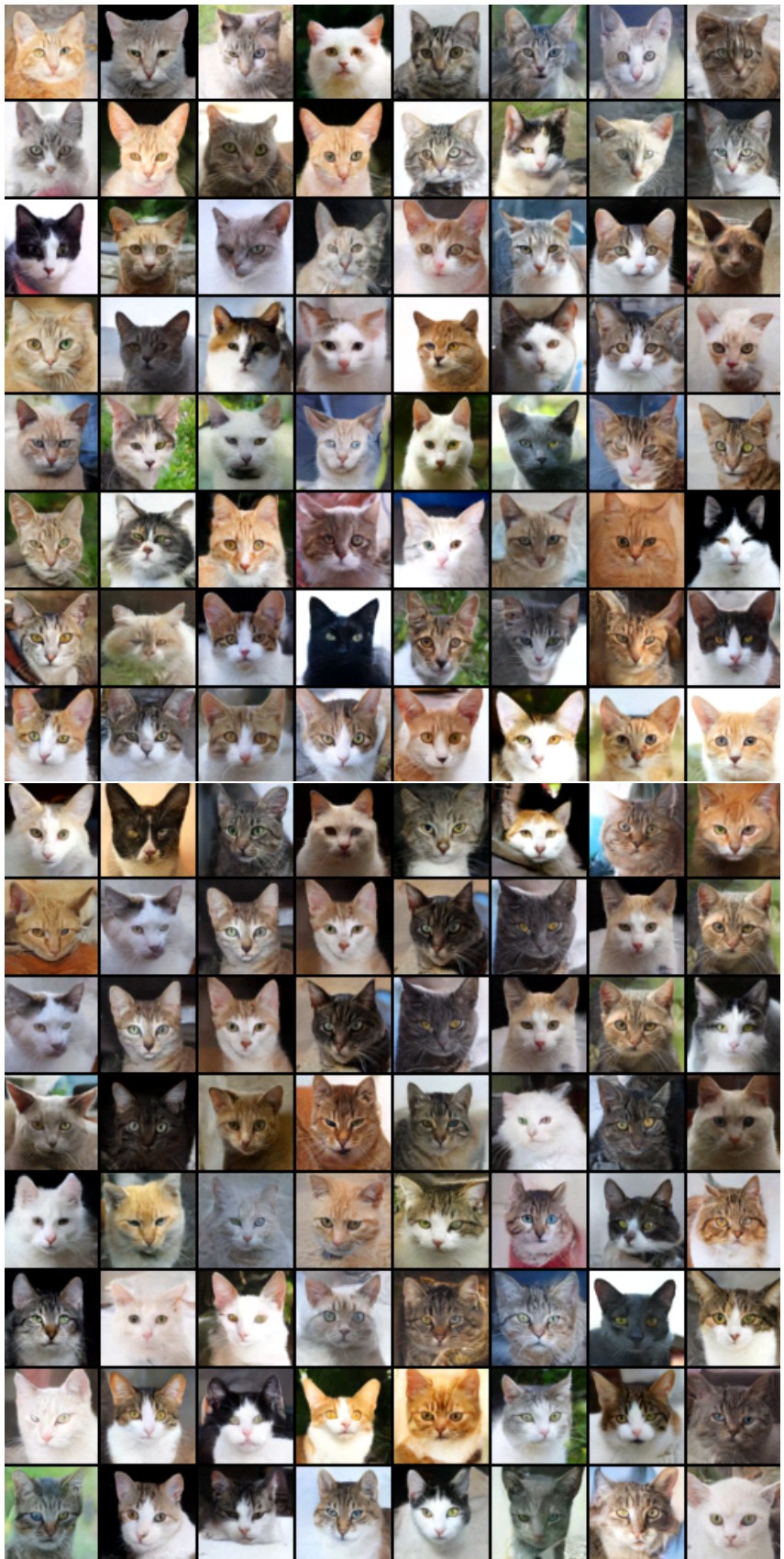

Figure 8: GP-GSBM on AFHQ dog → cat. Generated cat images starting from dog data samples.

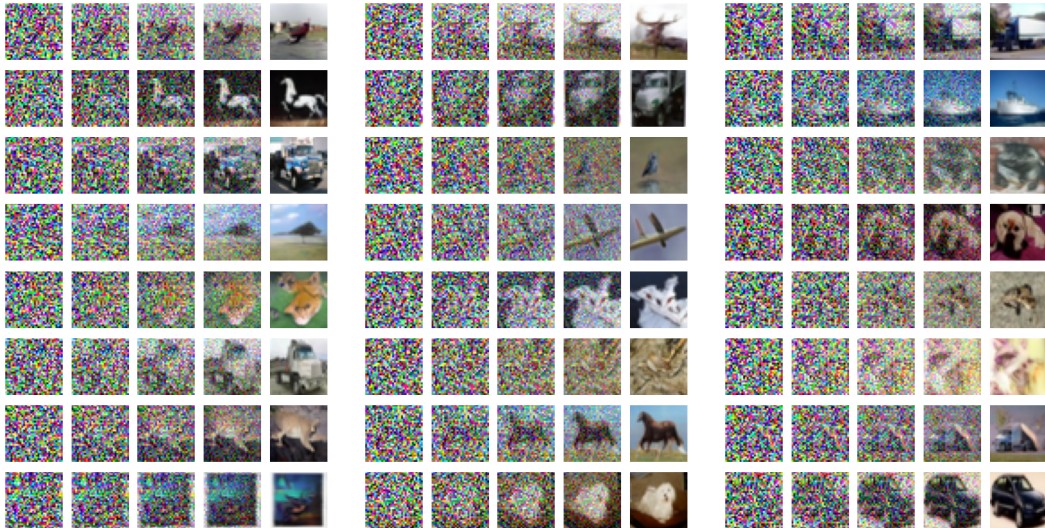

Figure 9: (CIFAR-10) SDE generation snapshots by our GP-GSBM. From the leftmost columns (samples from $\pi_0 = \mathcal{N}(0, I)$) to the generated images $\pi_1$ on the rightmost columns.

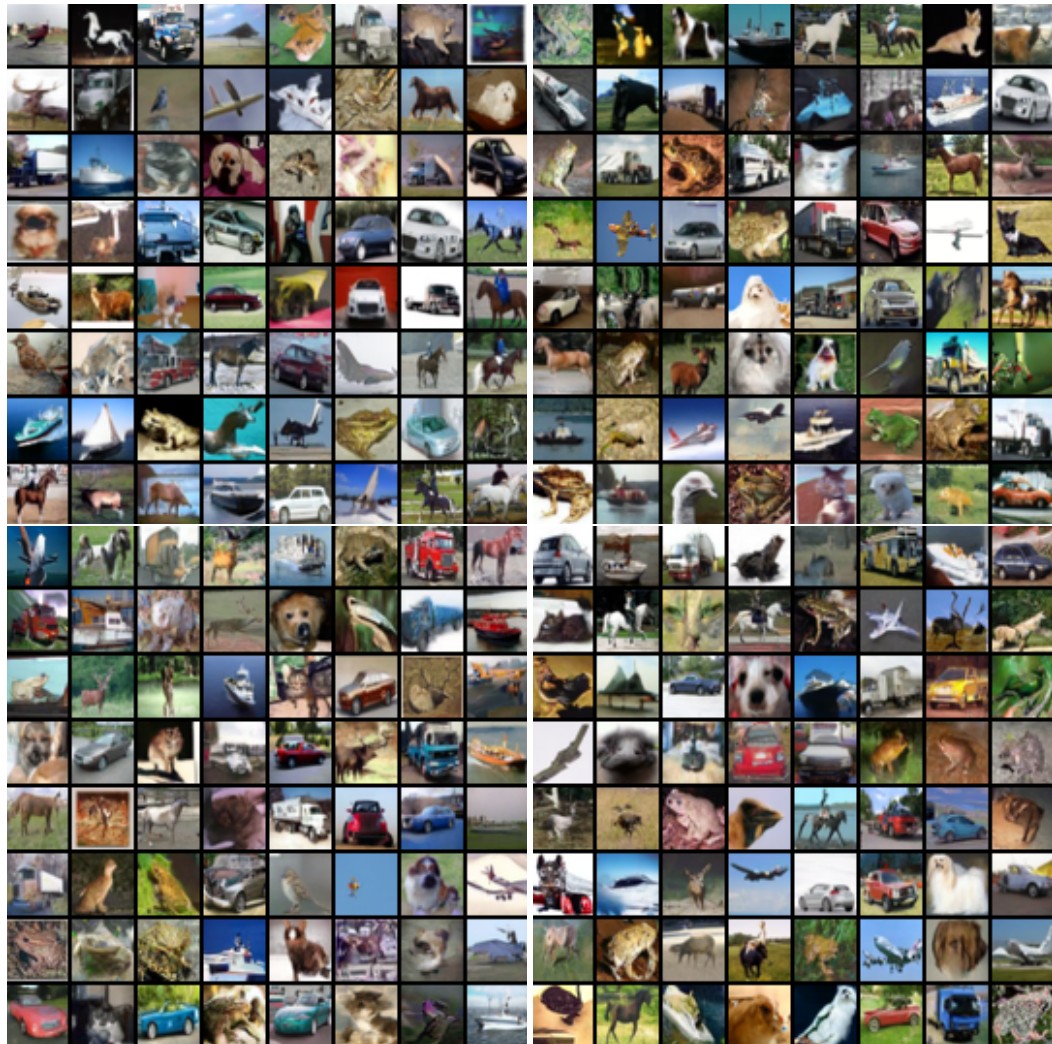

Figure 10: (CIFAR-10) Generated images from $\mathcal{N}(0, I)$ by our GP-GSBM.

# E    REVISION

This section contains added materials and new results from the reviews/rebuttal. We thank all the anonymous reviewers for their insightful and constructive comments and suggestions.

## E.1    TOY (THOUGHT) EXPERIMENT: WHY GP-GSBM IS ROBUST COMPARED TO GSBM

In this section we provide the key intuition on our (Bayesian) GP path modeling for the generalized Schrödinger bridge problem, illustrating why our GP-GSBM is more robust to noise than the (deterministic) GSBM (Liu et al., 2024) algorithm.

To focus on the core insight, we consider a highly simplified scenario, in particular, we make the following simplifications: i) the CondSOC cost (or the negative log-likelihood score in our Bayesian model) in (7) is queried from a black box oracle, meaning that it returns (stochastic) cost for a path query, ii) the posterior inference in our GP-GSBM is done exactly without resorting to approximation strategies like the sparse variational GP. For the former, although the CondSOC cost is composed of the kinetic cost and the problem-specific cost $V_t(x)$, we treat it as a single number retrieved from the oracle for simplicity.

In the 2D state space, we have two end points $x_0 = (-1, 0)$ and $x_1 = (+1, 0)$ sampled from the coupling distribution $Q(x_0, x_1)$. Let's consider $N = 1001$ paths connecting the two end points whose trajectories are given by parabolic functions $Y = a(X^2 - 1)$, that is, $[x_t]_2 = a([x_t]_1^2 - 1)$ for $a = i/500$, $i \in \{0, \pm 1, \ldots, \pm 500\}$ where $x_t = ([x_t]_1, [x_t]_2)$. They are visualized in Fig. 11. We assume that only these $N$ paths incur finite $(< \infty)$ cost (to be defined shortly), whereas any other path connecting $x_0$ and $x_1$ incurs $+\infty$ CondSOC cost.

Among these $N = 1001$ paths, let's say all paths incur CondSOC cost $+1$ except for the horizontal line path $Y = 0$ (i.e., $a = 0$, shown as red in Fig. 11) that incurs $1 - \epsilon$ where $\epsilon = 0.001$ most of the time. Specifically, the CondSOC cost for the horizontal line path is stochastic/noisy defined as:

$$J_{Y=0} = \begin{cases} 1 - \epsilon & \text{with prob. } 0.999 \\ 10^6 & \text{with prob. } 0.001 \end{cases} \tag{121}$$

Whereas the other 1000 paths incur cost $+1$ with probability 1, the path $Y = 0$ gives us slightly lower cost $1 - \epsilon$ most of the time, but can have the rare event of incurring extremely high cost $10^6$.

Now we are going to train the GSBM model and our GP-GSBM model, but let's assume that this rare event of the horizontal line path $Y = 0$ was not observed in the training data due to its rare event probability (0.001). We also note that due to the finite number of distinct paths with finite $(< \infty)$ cost, the optimal variance of the pinned marginal paths $P(x_t|x_0, x_1)$ for any sensible model must be 0 (i.e., $\gamma_t = 0$), and hence we only focus on the mean $\mu_t$.

First, GSBM is a deterministic point estimator, and thus it will learn the horizontal line $Y = 0$ path as its optimal path since it incurs slightly lower cost than any other 1000 paths, and we didn't observe the rare event in the training data. We can compute GSBM's (population) expected cost at test time (denoted by *risk*), which is

$$\text{Risk}_{\text{GSBM}} = \mathbb{E}_{noise}[J_{Y=0}] = (1 - \epsilon) \cdot 0.999 + 10^6 \cdot 0.001 \approx 1001 \tag{122}$$

Apart from GSBM's deterministic solution that only finds a point estimate (a single path) and exclude all the other possibility, the posterior in our Bayesian model offers us a distribution over a set of plausible paths under the given observations (observed costs). When we consider a very flat GP prior (i.e., nearly uniform prior over those $N$ paths), and if we use the temperature $\tau = 1$ for the likelihood model, the posterior will be proportional to $e^J$, more specifically,

$$\mathcal{P}_{post}(path) = \begin{cases} \frac{e^{-1+\epsilon}}{(N-1)e^{-1}+e^{-1+\epsilon}} & \text{if } path = \text{"}Y\!=\!0\text{"} \\ \frac{e^{-1}}{(N-1)e^{-1}+e^{-1+\epsilon}} & \text{if } path = \text{"}Y\!=\!a(X^2\!-\!1)\text{" for } a \in \left\{ \pm\frac{1}{500}, \ldots, \pm\frac{500}{500} \right\} \\ 0 & \text{otherwise} \end{cases} \tag{123}$$

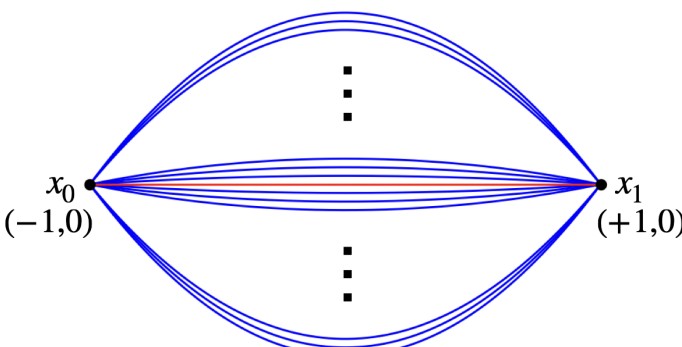

Figure 11: Toy (thought) experiment setup. $N = 1001$ paths connecting $x_0 = (-1, 0)$ and $x_1 = (+1, 0)$ are shown, which incur finite ($< \infty$) CondSOC costs. The horizontal line path $Y = 0$ (shown as red) incurs stochastic noisy cost $J_{Y=0}$ as defined in (121). The rest 1000 paths (blue) incur fixed cost $J = 1$. The GSBM (deterministic) algorithm finds the red $Y = 0$ path as a point estimate, which yields the risk $\approx 1001$. Our (Bayesian) GP-GSBM learns the near-uniform posterior distribution over those 1001 paths as in (123), achieving the risk $\approx 2$.

The risk of GP-GSBM is then computed as:

$$\text{Risk}_{\text{GP-GSBM}} = \mathbb{E}_{\mathcal{P}_{post}(path)}[\mathbb{E}_{noise}[J_{path}]] \tag{124}$$

$$= \frac{e^{-1+\epsilon}}{(N-1)e^{-1} + e^{-1+\epsilon}} \cdot ((1-\epsilon) \cdot 0.999 + 10^6 \cdot 0.001) + \frac{(N-1) \cdot e^{-1}}{(N-1)e^{-1} + e^{-1+\epsilon}} \cdot 1 \approx 2$$

This thought experiment, albeit quite simplified, provides the key intuition on why our Bayesian (GP) formulation for the generalized Schrödinger bridge problem is more robust to noise than (deterministic) GSBM (Liu et al., 2024).

## E.2 BACKGROUND ON GAUSSIAN PROCESSES (GP)

The GP models are special Bayesian models where we impose a prior distribution over a function, and the distribution has the property that for any input index set, the function values on the set follow Gaussian. Specifically, we consider the latent function $f : \mathcal{X} \to \mathbb{R}$ where $f(x)$, the function value at input point $x \in \mathcal{X}$, is a latent variable. The Gaussian process is characterized by the mean function $m : \mathcal{X} \to \mathbb{R}$ and the positive definite covariance (or kernel) function $k : \mathcal{X} \times \mathcal{X} \to \mathbb{R}$, defined as:

$$\mathbb{E}[f(x)] = m(x), \quad \text{Cov}(f(x), f(x')) = k(x, x') \tag{125}$$

for any $x, x' \in \mathcal{X}$. For any input set $X = \{x_1, \ldots, x_n\}$, the latent variables $(f(x_1), \ldots, f(x_n))$ jointly follow the multivariate Gaussian with mean $(m(x_1), \ldots, m(x_n))$ and the covariance matrix $K(X)$ whose $(i, j)$ entry is $k(x_i, x_j)$. The GP prior distribution $P(f(\cdot))$ is often denoted by $\mathcal{GP}(f(\cdot); m(\cdot), k(\cdot, \cdot))$, and we used this notation in our main paper.

Once we impose a GP prior on a latent function $f(\cdot)$, and once we have a likelihood model $P(\mathcal{D}|f(\cdot))$ that defines the likelihood of the observation (or observed data) $\mathcal{D}$ with respect to the function $f(\cdot)$, we can follow the Bayes rule to infer the posterior distribution over the functions. That is,

$$P(f(\cdot)|\mathcal{D}) \propto \mathcal{GP}(f(\cdot)) \cdot P(\mathcal{D}|f(\cdot)) \tag{126}$$

Note that there is no restriction on how the likelihood model $P(\mathcal{D}|f(\cdot))$ is defined. Each observed instance in $\mathcal{D}$ can be dependent on some or all of the function values $\{f(x)\}_{x \in \mathcal{A}}$ for some $\mathcal{A} \subseteq \mathcal{X}$. For instance, in the traditional GP regression (Rasmussen & Williams, 2006) where we model each input/output pair as $y = f(x) + \epsilon$ for some noise $\epsilon$, the likelihood of each observed instance $(x, y) \in \mathcal{D}$ depends on $f(x)$ at the point $x$ alone. However, for the CondSOC-based likelihood model (7) and (9) in our GP-GSBM, we have two types of latent functions $\mu_\bullet$ (or $\mu(\cdot)$) and $\gamma_\bullet$ (or $\gamma(\cdot)$), and the likelihood at each time $t$ depends on not just $(\mu(t), \gamma(t))$, but also those in their infinitesimal neighborhood $(\mu(t + \Delta t), \gamma(t + \Delta t))$ with $\Delta t \to 0$. This is because of the term $\alpha_t$ defined in (5).

Except for a few special cases (e.g., GP regression with Gaussian noise), most likelihood models in practice including the one in our GP-GSBM, lead to non-GP posterior $P(f(\cdot)|\mathcal{D})$. For tractable

solutions, one has to resort to approximate GP inference algorithms (Rasmussen & Williams, 2006). In this paper, we adopted the sparse variational free-energy approximate inference algorithm which is known to take various advantages (Titsias, 2009; Dezfouli & Bonilla, 2015; Matthews et al., 2016; Bauer et al., 2016).

### E.3 ENLARGED VERSION OF FIG. 1

In Fig. 12, we show an enlarged version of the images in Fig. 1 for better readability.

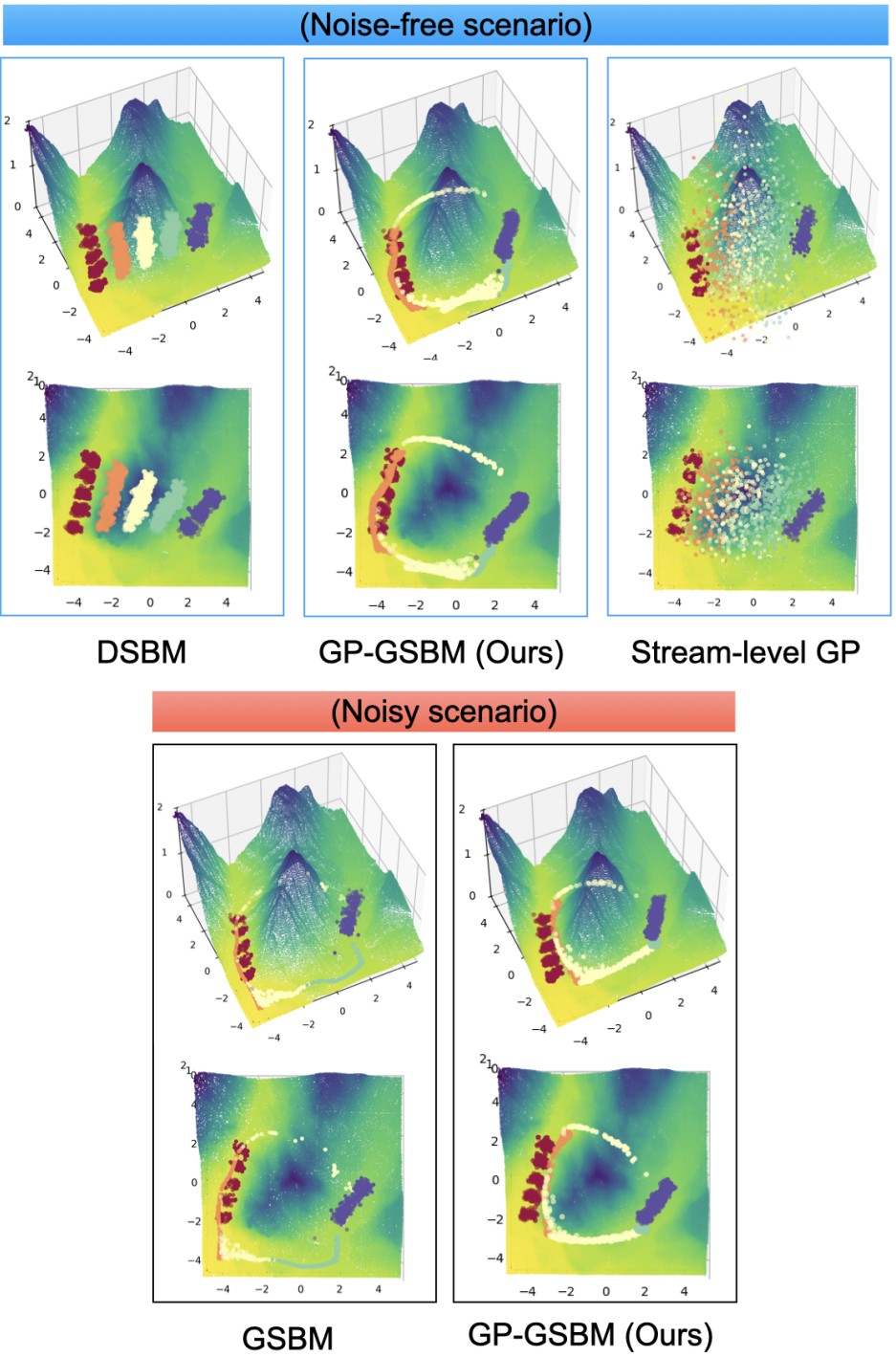

Figure 12: LiDAR path samples for different models in 3D (top rows) and 2D (bottom) views. Starting from the samples of $\pi_0$ (red points on the left), we generate samples for $\pi_1$ as blue points on the right. Path samples are visualized in 5 uniform time points from $t = 0$ (red) to $t = 1$ (blue).

