# OpenReview forum: "Robust Generalized Schr\"{o}dinger Bridge via Sparse Variational Gaussian Processes"
_ICLR.cc/2026/Conference — ICLR 2026 Poster_

### Official Review · Reviewer_PMLn · 2025-10-29

**Soundness:** 3
**Presentation:** 3
**Contribution:** 3
**Rating:** 6
**Confidence:** 3

**Summary:**

This paper proposes a GP prior GSBM algorithm for robust generalization. This allows a sparse variational free energy formulation that ensures effective posterior path measures using GP-based mean and covariance derivation on pinned marginals. The noticeable benefits are robustness in noisy scenarios and generality for various SDE settings. The authors presented various experiments demonstrating its effectiveness.

**Strengths:**

* The proposed method and ELBO optimization method can be applied in various circumstance.
* The overall manuscript is clearly written.
* The authors put considerable effort into experiments, and they successfully found the method's strength in robustness.

**Weaknesses:**

* Although it is referred to as generalization, I think the GP prior is another constraint or additional premise on SB that would only favor some portion of SDE problems in theory. I think a GP-based formulation can be overly restrictive for very high-dimensional modalities with additional geometric constraints that have branching paths as solutions.
* The precise notion of robustness is unclear, and why the GP prior recovers the "original" solution amidst noise is black-boxed.
* There is a room for further theoretical discussion on the GP prior which might benefit the clarity manuscript.

**Questions:**

.

---

> ### Author Response · Authors · 2025-11-20
> **Response to Reviewer PMLn**
>
> Thank you for your valuable feedback!
>
> > **1. GP-based formulation can be another constraint or restrictive.**
>
> We think that it is not correct in general to say that imposing a GP prior restricts the solution space, and we want to argue this from the Bayesian perspective and in terms of model selection.
>
> Our GP-GSBM is a Bayesian approach where we basically deal with all possible solutions (in this case, all possible pinned marginal path distributions). The GP prior is a way to express the preference over different solutions, where this preference is a priori, i.e., before seeing any observations. After seeing observations, then we amend the preference (posterior) by incorporating likelihood scores of different solutions in conjunction with their prior preferences.
>
> On the other hand, the previous work GSBM is non-Bayesian. But it can be seen as adopting a fixed special *uninformative* prior, meaning that all solutions are equally preferable a priori, and the final solution is solely determined by the likelihood score (the CondSoc objective value).
>
> In our GP-GSBM, we have the freedom to choose this prior preference. Specifically, we can control the prior preference through the GP kernel covariance function. In the extreme case, by choosing a very large covariance function, we can effectively impose an uninformative prior, which matches the GSBM solution in some sense. But if we want to put higher priority on more smooth solutions, we can use other relevant kernel functions. The model selection (empirical Bayes) is a principled way to choose the most appropriate kernel function based on the data/observation. In other words, which solution class is the most appropriate can be determined in a principled data-driven way within our Bayesian framework.
>
> Regarding the comment on restrictiveness of GP-based formulation:
> The Gaussian inducing variable modeling in our GP posterior approximation is quite flexible, and does not restrict the expressibility at least compared to GSBM’s linear spline path modeling. For further details, please see our responses to the second question from Reviewer [Jgjn](https://openreview.net/forum?id=3a2QuEzveq&noteId=Eohlp50Wzu).
>
> We hope this helps answer your question.
>
>
> > **2. Precise notion of robustness.**
>
> In the revised paper, we have added a toy (thought) experiment section, which hopefully can give some better intuition and notion of robustness, and why our approach is better than GSBM under noisy cost scenarios. Please see Appendix E.1 for details. Albeit quite simplified, it provides the key intuition on why our Bayesian (GP) formulation for the generalized Schr\"{o}dinger bridge problem is more robust to noise than (deterministic) GSBM.
>
>
> > **3. Further theoretical discussion on the GP prior.**
>
> If the reviewer meant inclusion of some theoretical background on GP prior, we have added some background material and discussion on our GP prior in the revised paper. Please see Appendix E.2 of the revised manuscript. Otherwise, please excuse what we didn’t understand, and we would appreciate it if the reviewer clarifies the question further.

---

### Official Review · Reviewer_ofu8 · 2025-10-31

**Soundness:** 3
**Presentation:** 3
**Contribution:** 3
**Rating:** 8
**Confidence:** 2

**Summary:**

This paper extends the Schrödinger Bridge (SB) framework by introducing a robust generalized formulation that unifies diffusion-based generative modeling and SB matching under a common theoretical lens. The authors develop a new objective function incorporating robustness to model mismatch and derive corresponding training dynamics. The work provides both theoretical guarantees and empirical validation on complex datasets, aiming to improve the stability and expressiveness of SB-based generative models. But to be honest, this paper is too mathematically dense and is hard for me to follow.

**Strengths:**

The paper tackles an ambitious and mathematically sophisticated problem, presenting a unifying and robust extension to the Schrödinger Bridge framework. The theoretical development appears rigorous, and the results—both in proofs and experiments—suggest meaningful advances in understanding and improving bridge-based generative modeling. The connection drawn between generalized SB matching and diffusion-based models is conceptually strong and of potential significance to the community.

**Weaknesses:**

The paper is mathematically dense and can be challenging to follow, even for readers with backgrounds about SB (not GSB). Some of the derivations could benefit from more intuition or interpretive commentary. It is also difficult, without deep verification of each step, to fully assess the soundness of the theoretical proofs.

**Questions:**

Could the authors provide a more intuitive explanation or diagram illustrating how the generalized robust SB formulation differs from the classical one, and how robustness manifests in practice?

---

> ### Author Response · Authors · 2025-11-20
> **Response to Reviewer ofu8**
>
> Thank you for your valuable feedback!
>
> > **1. More intuitive explanation on the proposed approach regarding the robustness in GSB.**
>
> In the revised paper, we have added more intuitive illustrations for the underlying key idea of our approach regarding robustness. Please see Appendix E.1 for the toy (thought) experiment. Albeit quite simplified, it provides the key intuition on why our Bayesian (GP) formulation for the generalized Schr\"{o}dinger bridge problem is more robust to noise than (deterministic) GSBM.

---

### Official Review · Reviewer_Jgjn · 2025-11-01

**Soundness:** 2
**Presentation:** 3
**Contribution:** 2
**Rating:** 4
**Confidence:** 4

**Summary:**

This study proposes a variational solution that applies a Gaussian Process (GP) to efficiently solve the existing Generalized Schrödinger Bridge (GSB) problem. To define the Gaussian process, Gaussian distributions are first established based on the data from two domains, $x_0$ and $x_1$. A GP prior is then constructed between them, and learning is conducted by optimizing the Evidence Lower Bound (ELBO) over intermediate timesteps. Using this approach, the proposed method achieves superior performance compared to existing approaches.

**Strengths:**

- Simplification through a variational approach: The proposed method applies a Gaussian Process (GP) to the GSB problem, allowing convenient formulation of intermediate distributions through a simple prior. This variational perspective simplifies the overall problem structure while maintaining flexibility.

- Visualization of results: The paper thoroughly validates the proposed methodology through diverse visual analyses presented in both the main manuscript and the appendix. These visualizations effectively demonstrate the distinctions and advantages of the proposed approach.

**Weaknesses:**

### Major Weaknesses
- Time Complexity: As also mentioned by the authors, the proposed method has significantly higher time complexity compared to the original GSBM. According to Table 5 in the Appendix (Page 26), the method additionally requires $O(n^3)$ time. A strategy to further reduce the sampling time is needed. Considering the current computational cost, the performance improvement over GSBM does not appear dramatic.

- “Sparse” Variational Gaussian Process: The proposed method constructs a flow by interpolating simple single Gaussian kernels defined on two domains, $x_0$ and $x_1$. While this simplifies the problem, it does not fully capture the global distribution, making the selection of data pairs highly influential to overall performance.

- Uncertain Performance Difference: The proposed approach modifies the GSBM structure only by formulating $V_t$ as a GP problem. Based on the reported results, including those in the Appendix, the improvement over GSBM appears only incremental. It would strengthen the paper to include concrete experimental cases where GP-GSBM successfully solves problems that GSBM fails to address.

### Minor Weaknesses

- Choice of Primary Area: The proposed method utilizes a Gaussian Process (GP) as one possible solution to the GSB problem. In my opinion, this work may fit more appropriately under the area of generative modeling rather than GP itself.

- Formatting in the Related Work Section: It might be better to remove quotation marks from the paragraph titles in the Related Work section for cleaner presentation.

**Questions:**

See Weaknesses Section.

---

> ### Author Response · Authors · 2025-11-20
> **Response to Reviewer Jgjn**
>
> Thank you for your valuable feedback!
>
> > **1. (major) Time complexity.**
>
> As we mentioned in Appendix B.2, the additional cubic time can be amortized, ie, the kernel inversion is done once before the ELBO learning starts, then reused thereafter, if we do not perform gradient-based model selection (kernel hyperparameter learning). But if we have to do gradient-based model selection during the ELBO learning, the current version has this drawback of cubic cost at each iteration, as we also stated in the limitation section.
>
> We haven’t dealt with this issue in a principled manner in the current work since the current paper is already dense. But our current workaround is doing model selection by grid search (as stated in L:1433~1438): for a discrete set of candidate kernel hyperparameters, we train our model with each candidate, and choose the one with the highest ELBO score. Considering that the candidate set is typically small (<10 candidates) in practice, this can reduce time considerably.
>
> Regarding reviewer’s suggestion, some more principled sampling strategy can be considered, including the GP decomposition method (Wilson et al. 2020), which can be further pursued as future research.
>
> *(Wilson et al. 2020) Efficiently Sampling Functions from Gaussian Process Posteriors, J. T. Wilson, V. Borovitskiy, A. Terenin, P. Mostowsky, M. P. Deisenroth, ICML 2020.*
>
>
> > **2. (major) Sparse Variational GP: Simple single Gaussians.**
>
> If the reviewer’s question is about the expressibility of the Gaussian modeling of the inducing variables in Eq.(15), we respond to it as follows. Otherwise, please excuse what we didn't understand, and we'd appreciate it if the reviewer explains the question in greater detail.
>
> The Gaussian inducing variable modeling is quite flexible, and does not restrict the expressibility at least compared to GSBM’s linear spline path modeling. To see this, we can look at the functional form of the induced posterior mean and covariance functions in Eq.(17-20). For instance, in Eq.(17), the mean function for $\mu$ is parametrized by $C$ that has as many free parameters as the spline knot free parameters in GSBM. Unlike GSBM’s spline interpolation where the value at time $t$ depends only on two nearby knot points, we have a form where *all* entries of $C$ can affect the value at $t$, determined by the coefficient matrix  $L_{t,Z}L_{Z,Z}^{-1}$ and also the bias term containing $M_t$. In contrast to our full coefficient matrix, the linear spline in GSBM can be seen as having a special bidiagonal coefficient matrix leading to only 0-curvature curves (ie, lines). Hence our posterior form can give us smooth and highly flexible curve modeling.
>
> Regarding the selection of data pairs, just a friendly reminder that pairs $(x_0, x_1)$ are selected by sampling from the coupling distribution $Q(x_0,x_1)$. Please note that this coupling distribution is updated at each iteration (Eq.(4) updates $Q$ with the updated drift network in step 3 of Alg. 1) following the DSBM/IMF principle, similarly as the GSBM paper (and more specifically, within our unified bridge framework). Another thing to note is that we perform the GP posterior inference for each pair $(x_0, x_1)$ sampled from coupling $Q(x_0, x_1)$, just like they solved the CondSOC optimization for each pair $(x_0, x_1)$ in the GSBM paper.
>
> We hope this helps answer your question.
>
>
> > **3.(major) Uncertain performance difference.**
>
> To highlight the key benefit over GSBM and illustrate our main intuition regarding robustness, we have included a new toy experiment (or thought experiment) in the revised paper. Please see Appendix E.1. This thought experiment, albeit quite simplified, provides the key intuition on why our Bayesian (GP) formulation for the generalized Schr\"{o}dinger bridge problem is more robust to noise than (deterministic) GSBM.
>
> Regarding the improvement over GSBM:
> In the original paper, we have reported the standard deviations for the reported objective values for LiDAR, Stunnel and GMM experiments in Table 1 and 2. At least for the noisy/uncertain scenarios that our approach is targeting particularly, we have results statistically significantly better than GSBM. To clarify it, we have conducted the paired $t$-test between GSBM and ours, and the following table shows the $p$ values.
>
> |   | GSBM | GP-GSBM | $p$-value |
> |----|----|----|----|
> | LiDAR | $8506.1 \pm 65.6$ | $8300 \pm 67.6$ | $0.0002$ |
> | Stunnel | $502.2 \pm 1.76$ | $452.3 \pm 1.78$ | $< 10^{-6}$ |
> | GMM | $101.6 \pm 1.02$ | $89.2 \pm 0.62$ | $< 10^{-6}$ |
>
> All $p$ values are less than 0.01, implying that the improvement is statistically significant.
>
> *(Continued in the next comment box)*

---

> > ### Author Response · Authors · 2025-11-20
> > **(Continued from the previous comment box)**
> >
> > > **4. (minor) Choice of primary area.**
> >
> > We also deliberated considerably. A key rationale for selecting GP as a primary area is that the GSB problem offers an interesting and topical application, allowing GP modeling to capture uncertainty and find robust solutions. But what the reviewer suggested also makes very good sense. We will keep thinking about it.
> >
> >
> > > **5. (minor) Formatting in the Related Work section.**
> >
> > We have revised it as the reviewer suggested.

---

### Official Review · Reviewer_gihK · 2025-11-03

**Soundness:** 3
**Presentation:** 3
**Contribution:** 3
**Rating:** 8
**Confidence:** 2

**Summary:**

This manuscript tackles the generalized Schrodinger Bridge (GSB) problem that incorporates an extra stage cost function on top of the original SB problem. Given that the previous efficient GSB solver only offers a point estimate of the path measure, the authors propose to treat the GSB as a posterior inference problem. Specifically, they treat the GSB objective as a stochastic likelihood function and leverage a Gaussian path prior, and aim to estimate the posterior distribution. To address the intractable posterior, the authors propose to leverage fully factorized variational inference, where the variational posteriors are tractable GPs. Empirical evaluation demonstrates that the proposed solution has competitive performance with the GSBM baseline and superior performance on the scenarios where there is uncertainty or under a proxy cost function.

**Strengths:**

* The writing is clear, with good contextualization of related works.
* The motivation of this paper is good.
* The proposed technique is sound, with well-executed experiments.

**Weaknesses:**

Overall, I have no major concerns about this manuscript.

### Minor
* The implications or limitations of some design choices should be highlighted. For example, will the independent prior (regularization) be suitable for all applications? The same question applies to the homogeneous kernel function across dims (though this is common) and the form of the variational posterior. However, adding complexity to these designs may further increase computational cost.
* The figure's resolution can be improved. The current form is somewhat hard to read when the paper is printed.

**Questions:**

See weaknesses.

---

> ### Author Response · Authors · 2025-11-20
> **Response to Reviewer gihK**
>
> Thank you for your valuable feedback!
>
> > **1. (minor) Implication or limitations of some design choices.**
>
> Yes, as the reviewer mentioned, more flexible modeling (e.g., dimension-wise different kernel functions) would increase the computational overhead considerably. So, it is mainly for computational tractability as we stated in the paragraph before Eq.(10).
>
>
> > **2. (minor) Resolutions of figures.**
>
> We guess that the reviewer is referring to Fig.1 and Fig.2. We have checked the resolutions of these images, but they seem to be already high enough. So, to resolve the reviewer’s concern regarding readability, we have included the enlarged version of Fig.1 in Appendix E.3  (Fig.12) in the revised paper. (For Fig.2, the original paper already includes the enlarged version in Appendix D.1). We hope this reads better.

---

### Author Response · Authors · 2025-11-20
**Revised manuscript with additional experiments and clarifications**

We thank all reviewers for their insightful and constructive comments/questions.

In response to reviewers' comments, in Appendix E of the revised manuscript, we have included additional experiments and clarifications, especially, **toy/thought experiment (Appendix E.1)** and **background and more discussion on GP priors (Appendix E.2)**.

Our responses to reviewers' individual comments/questions are placed after each review.

---

### Meta-Review · Area_Chair_zRAP · 2025-12-30

**Summary:**

This paper proposes a technique for solving (generalized) Schrodinger Bridge problems via sparse variational GPs. Reviewers universally appreciate the conceptual novelty of the proposed approach and comment that the experiments are well-executed and generally convincing.

Reviewers comment on the following weaknesses:
- [W1] Clarity of presentation (gihK, ofu8)
- [W2] Questions about particular design choices (gihK, PMLn)
- [W3] Concerns about computational efficiency (Jqjn)

**Reviewer Concerns:**

- [W1] The authors take several steps in the rebuttal to improve the clarity and presentation of the work.
- [W2] The rebuttal largely addresses these concerns.
- [W3] The authors explicitly acknowledge this as a potential limitation of the methodology. However, it does not seem prohibitively expensive, and the rebuttal provides several ideas or directions for improving upon this limitation. Thus, this weakness seems largely addressed.


Overall, this paper seems to be a novel and well-executed contribution on a timely and relevant problem, and thus I recommend acceptance.

**Reviewer Scores:**

- gihK, ofu8, PMLn already give high scores and seem unlikely to further increase their ratings.
- Jqjn already has high confidence, but may have increased their score slightly as the rebuttal addresses the major concerns raised by the reviewer.

---

### Decision · Program_Chairs · 2026-01-26

Accept (Poster)